# The Cumulus And Stratocumulus CloudSat-CALIPSO Dataset (CASCCAD)

Grégory Cesana[1,2], Anthony D. Del Genio[2] and Hélène Chepfer[3,4]

[1] Columbia University, Center for Climate Systems Research, Earth Institute, New York, NY, USA
[2] NASA Goddard Institute for Space Studies, New York, NY, USA
[3] LMD/IPSL, Sorbonne Université, Paris, France
[4] LMD/IPSL, CNRS, École Polytechnique, Palaiseau, France

*Correspondence to*: Gregory V. Cesana (Gregory.cesana@columbia.edu)

To submit to Earth System Science Data (copernicus open discussion)

**Abstract.**

Low clouds continue to contribute greatly to the uncertainty in cloud feedback estimates. Depending on whether a region is dominated by cumulus (Cu) or stratocumulus (Sc) clouds, the interannual low-cloud feedback is somewhat different in both space-borne and large eddy simulation studies. Therefore, simulating the correct amount and variation of the Cu and Sc cloud distributions could be crucial to predict future cloud feedbacks. Here we document spatial distributions and profiles of Sc and Cu clouds derived from Cloud-Aerosols Lidar and Infrared Pathfinder Satellite Observations (CALIPSO) and CloudSat measurements. For this purpose, we create a new dataset called the Cumulus And Stratocumulus CloudSat-CAlipso Dataset (CASCCAD), which identifies Sc, broken Sc, Cu under Sc, Cu with stratiform outflow and Cu. To separate the Cu from Sc, we design an original method based on the cloud height, horizontal extent, vertical variability and horizontal continuity, which is separately applied to both CALIPSO and combined CloudSat-CALIPSO observations. First, the choice of parameters used in the discrimination algorithm is investigated and validated in selected Cu, Sc and Sc-Cu transition case studies. Then, the global statistics are compared against those from existing passive and active-sensor satellite observations. Our results indicate that the cloud optical thickness –as used in passive-sensor observations– is not a sufficient parameter to discriminate Cu from Sc clouds, in agreement with previous literature. Using clustering-derived datasets show better results although one cannot completely separate cloud types with such an approach. On the contrary, classifying Cu and Sc clouds and the transition between them based on their geometrical shape and spatial heterogeneity leads to spatial distributions consistent with prior knowledge of these clouds, from ground-based, ship-based and field campaigns. Furthermore, we show that our method improves existing Sc-Cu classifications by using additional information on cloud height and vertical cloud fraction variation. Finally, the CASCCAD datasets provide a basis to evaluate shallow convection and stratocumulus clouds on a global scale in climate models and potentially improve our understanding of low-level cloud feedbacks. The CASCCAD dataset (Cesana,

2019, DOI: http://doi.org/10.5281/zenodo.2667637) is available on the Goddard Institute for Space Studies (GISS) website at https://data.giss.nasa.gov/clouds/casccad/ and on the zenodo website at https://zenodo.org/record/2667637.

**1 Introduction**

Whether clouds will amplify or dampen global warming, referred to as cloud feedbacks, continues to be a dominant source of uncertainty in future climate projections, for which low clouds over the tropics and at mid-latitudes contribute up to 50 % in recent generations of the Coupled Model Intercomparison Project (CMIP) models (Zelinka et al., 2016). Low clouds (i.e., cloud top height lower than ~ 3 km) may be separated into two main categories: stratocumulus (Sc, including stratus clouds)

and cumulus (Cu). Driven by radiative cooling, Sc clouds cap the planetary boundary layer over cool oceans in conditions with a strong cloud top inversion, mostly off the western coasts of continents (e.g., Klein and Hartmann, 2002). They are typically hundreds of meters thick with a large horizontal extent, which can be either homogeneous (in decks) or heterogenous (open and closed cells), and a stable cloud-top height. Due to their large cloud cover, these clouds strongly reflect shortwave (SW) radiation and contribute to surface cooling. As the ocean warms up further west in the trade-wind regions, the latent heat flux

increases and the convection becomes surface driven, therefore favoring breaking up of Sc and the subsequent formation of Cu clouds (Albrecht et al., 2019; Wyant et al., 1997). These clouds are horizontally limited and scattered – i.e., with a modest cloud cover – and their tops can rise above the PBL. Since these low clouds are governed by distinct processes, they may respond differently to climate warming (e.g., Bretherton, 2015) and there is no fundamental reason to expect the two cloud types to exhibit the same feedback.

Idealized large-eddy simulation (LES) studies have partly supported the hypothesis of different cloud type responses to climate warming, i.e., a substantial decrease of Sc as opposed to a moderate decrease to no change for Cu, yet, the underlying processes remain poorly understood, particularly for trade-wind Cu (Bretherton, 2015). Furthermore, recent satellite-based studies have shown that the interannual low-cloud feedback in response to SST forcings is somewhat different depending on whether a

region is dominated by Cu or Sc clouds (Cesana et al., 2019a; McCoy et al., 2017). Therefore, simulating the right amount of Cu and Sc clouds could be crucial for models to reproduce the overall interannual low-cloud feedback as observed from space (Cesana et al., 2019a) and to predict future cloud feedbacks (Klein and Hall, 2015). Yet the relatively scarce amount of observations that fundamentally distinguish Sc and Cu clouds (mostly field campaigns and ground-based sites, i.e, Zhou et al., 2015; Rémillard et al., 2012) limits our ability to study the present-day global distribution of these clouds and their response

to surface warming, and hence to better constrain the climate models.

Estimating the global radiative impact of clouds on past, present and future climate continues to be a challenging question that requires observations of cloud macrophysical (e.g., height, spatial extent) and microphysical (e.g., phase, effective radius) properties on a global scale. Knowledge of the cloud type provides only a little leverage on determining their radiative properties, which may explain why cloud-type classification has received far less attention in the past. The first global-scale cloud-type observations were collected visually from land stations and ships in the 1950s and were subsequently compiled to make a coarse digital database in the late 1980s (Hahn et al., 1988) and were updated a couple of decades after (Hahn and Warren, 2007). A few years later, the first global-scale cloud type climatology derived from passive-sensor satellites emerged, based on cloud top pressure (CTP) and cloud optical thickness (COT) (Rossow and Schiffer, 1991, 1999). While very useful because of its long-time record, large spatial cover and finer resolution than Hahn et al. (1988), such datasets suffer from both methodology and instrumental limitations that make it difficult to fully discriminate Sc from Cu clouds. The CTP-COT method does not exploit information on the spatial shape of an individual cloud, that is to say its horizontal and vertical extent, and thus is not always accurate (Hahn et al., 2001; Pincus et al., 1999). Additionally, passive-sensor satellites do not describe the entire profile but only the uppermost layers, integrated over a height that is moreover hard to quantify with less confidence over land and in regions of strong inversions (Garay et al., 2008; Marchand et al., 2010). Thus, low cloud cover in regions of extensive higher clouds is underestimated. Finally, some instruments may not be well-suited for Cu cloud detection (Marchand et al., 2010).

By combining collocated observations from active CloudSat Cloud Profiling Radar (CPR, Stephens et al., 2002) and Cloud-Aerosols Lidar and Infrared Pathfinder Satellite Observations (CALIPSO; Winker et al., 2010) lidar and passive Moderate Resolution Imaging Spectroradiometer (MODIS; King et al., 2013) spectrometer, the 2B-CLDCLASS-LIDAR product (Sassen and Wang, 2008; Wang et al., 2013) addresses most of the above caveats. It classifies clouds into eight types based on cloud vertical and horizontal extent, reflectivity, precipitating state, temperature, height and brightness (Huang et al., 2015). However, this combined dataset is only available for a relatively short period of time (about 4.5 years), which influences statistical correlations between environment variables and cloud fraction (e.g., Cesana et al., 2019a; Klein et al., 2017). Although the radar-only product extends over a longer time-period (for daytime only, see section 2.2), the CPR is less sensitive to fractionated and thin shallow cumulus clouds than the CALIPSO lidar and its ground clutter prevents cloud detection below 1 km, which preclude the detection of a large amount of marine low-level clouds (mostly Sc and Cu clouds, Liu et al., 2016). Furthermore, the CPR horizontal resolution (~ 1.4 km x 1.7 km) is not ideal for shallow cumulus detection, which are typically smaller than 1 km (Rodts et al., 2003; Zhang and Klein, 2013). Therefore, creating a specific Sc-Cu cloud classification product based on CALIPSO observations only would allow one to overcome these issues although confining the analysis to regions with optically thin or no overlying high-clouds.

Here we propose to document spatial distributions and profiles of Sc, Cu and transitioning (i.e., broken Sc, Cu under Sc and Cu with stratiform outflow) clouds derived from CALIPSO measurements. To achieve this goal, we create an original method

based on the cloud height, horizontal extent, vertical variability and horizontal continuity, which can be applied separately to both CALIPSO and combined CloudSat-CALIPSO observations, referred to as the Cumulus And Stratocumulus CloudSat-CAlipso Dataset (CASCCAD, Cesana, 2019; DOI: http://doi.org/10.5281/zenodo.2667637). The datasets are presented in section 2. The sensitivity of the CASCCAD algorithm is assessed in section 3 and the global-scale results are then discussed

and compared to a subset of existing cloud-type datasets in section 4. Finally, section 5 summarizes the results.

## 2 Datasets

In this section, we describe the CALIPSO (GOCCP, section 2.1) and the combined CloudSat-CALIPSO (RL-GeoProf, section 2.2) observations that we use with the CASCCAD algorithm and the only other active-sensor cloud-type product available (2B-CLDCLASS-LIDAR, section 2.3). The CASCCAD algorithm (see full description in section 3) is designed so that it can

be applied to active-sensor level 2 observations (i.e., cloud mask at the orbital level) at their native resolution to generate level 2 Sc-Cu cloud mask profiles, which consist of along-track profiles of Sc, Cu and transitioning (i.e., broken Sc, Cu under Sc and Cu with stratiform outflow) cloud types. Here we separately apply CASCCAD to GOCCP and CloudSat-CALIPSO RL-GeoPRof. The resulting GOCCP- or CloudSat-CALIPSO-CASCCAD Cu-Sc profiles are then accumulated onto a 2.5˚x2.5˚ grid (2D cloud fraction) and over 40 480-m levels (3D cloud fraction) for each month to create level 3 files (i.e., cloud fraction

on the global scale). A detailed description of the content of both the GOCCP and RL-GeoProf CASCCAD level 3 files – averaged over the whole period of time, 2007 to 2016 for GOCCP and 2007 to 2010 for RL-GeoProf– are available on the CASCCAD website here: https://data.giss.nasa.gov/clouds/casccad/. To summarize, the distinct GOCCP and RL-GeoProf CASCCAD datasets contain 2D and 3D cloud fraction of all-low, Cu, Sc and transitioning (i.e., broken Sc, Cu under Sc and Cu with stratiform outflow) cloud type. Finally, the GOCCP and RL-GeoProf level 2 files (i.e., along-track Cu-Sc mask at

native resolution for each granule/orbit) are not yet available on the website (as of August 2019) but can be made available upon request.

## 2.1 GOCCP

The instantaneous cloud mask of the GCM-oriented CALIPSO Cloud Product (GOCCP) version 3.1.2 (Guzman et al., 2017)

is used by our discrimination algorithm (referred to as DA, described in section 3.2) to derive Sc-Cu cloud fraction statistics from 2007 to 2016 over a 2.5˚ grid and for 40 levels with 480 m spacing from 0 to 19.2 km. GOCCP (Chepfer et al., 2010) was developed to facilitate the evaluation of cloud properties in GCMs when combined with a lidar simulator (Chepfer et al., 2008) that uses the same cloud definitions, and ensures a consistent comparison between observations and simulations (Cesana and Waliser, 2016). The ratio of the Total Attenuation Backscatter signal (ATB) to the molecular ATB – so-called Scattering

Ratio (SR) – is computed from the level1B CALIPSO files for every 333 m-along-track-resolution near-nadir lidar profile for 480 m height intervals. This lidar-based quantity is a proxy of the presence of particulate matter in a layer. GOCCP uses a fixed SR threshold to detect clouds (SR>5), for either daytime or nighttime data, regardless of the vertical level. This threshold

allows the detection of thin cirrus cloud in the high-troposphere (McGill et al., 2007), hence the majority – if not all – of optically thicker PBL clouds except when masked by overlying high-clouds (e.g., the trade-wind regions). It also prevents most false detections of aerosol layers as being cloudy in the PBL (Chepfer et al., 2013). GOCCP has been validated against in situ (Cesana et al., 2016) and ground-based observations (Lacour et al., 2017). Caveats for this dataset are discussed in

5    Cesana et al. (2016) and in Cesana and Waliser (2016). Compared to GOCCP version 2.9 (Cesana et al., 2016), version 3.1.2 improves the detection of fully attenuated pixels by introducing a surface echo detection. When no lidar echo is detected, pixels below the lowest cloudy pixel are diagnosed as being fully attenuated and therefore not accounted for in the cloud fraction computation. This new feature reduces the underestimation of the cloud fraction underneath optically thick liquid-topped clouds in the lower troposphere (Cesana et al., 2016).

## 2.2 CloudSat-CALIPSO RL-GeoProf

Additionally, the Sc-Cu DA is applied to combined CloudSat-CALIPSO profiles (using the radar-lidar geometrical profile product [RL-GeoProf], Mace and Zhang, 2014; version R04) from 2007 to 2010 over a 2.5˚ grid and for 80 levels with 240 m spacing from 0 to 19.2 km. From its launch to early April 2011, CloudSat flew approximately 15 s ahead of CALIPSO making

it possible to observe the same scene from a lidar-radar perspective when using both the 2B-GEOPROF and 2B-GEOPROF-LIDAR products (Mace and Zhang, 2014). CloudSat experienced a severe anomaly in April 2011, which forced the satellite to leave the A-Train constellation before coming back in June 2012 in "Daylight Only Operation mode" (DO-Op). As a result, CloudSat-CALIPSO combined observations are only available for 4.5 years. The CPR cloud mask provides confidence levels for the cloud detection (Marchand et al., 2009). Following Mace and Zhang (2014), we chose the confidence level of 20 and

higher to characterize the presence of a cloud from the radar cloud mask. The CPR has a coarser horizontal (~ 1.4 km x 1.7 km) and vertical resolution (240 m) than the CALIPSO lidar ($\Delta z = 30$ m below 8 km and 60 m above 8 km for ~ 70 m footprint every 333 m). As a result, several lidar profiles fall within the CloudSat radar footprint most of the time. These are used to compute the lidar cloud fraction ($CF_{lidar}$, from 0 to 1) based on the CALIPSO Vertical Feature Mask (VFM) of the level2 5km CALIPSO files (Vaughan et al., 2009), at the CPR resolution in the 2B-GEOPROF-LIDAR files. While the $CF_{lidar}$ can

sometimes be lower than 0.5, we kept the 0.5 threshold to diagnose the presence of a cloud as in Mace and Zhang (2014) and Cesana et al. (2019b). Diagnosing a pixel as cloudy from values below 0.5 may result in an overestimate of the averaged cloud fraction when compared to ground-based measurements (Fig. S1 and Marchand et al., 2010).

## 2.3 2B-CLDCLASS-LIDAR

The 2B-CLDCLASS-LIDAR product R05 (Sassen and Wang, 2008; Wang et al., 2013; referred to as 2BCCL in the remainder of the manuscript) merges collocated observations from the CloudSat CPR, CALIPSO lidar and MODIS spectrometer to classify clouds into eight types based on several criteria: their vertical and horizontal extent, their precipitating state, their temperature, and their radiance. Although eight cloud types are available in this dataset (Deep convective, Cirrus, Nimbostratus, Altostratus, Altocumulus, Cumulus including fair-weather and congestus, Stratus and Stratocumulus), we only

focus on the Cu, stratus (St) and Sc cloud-types. The St and Sc cloud-types are combined into a single category referred to as Sc for consistency with the Sc-Cu discrimination algorithm, which does not differentiate these two categories. In addition, the Sc and St clouds are particularly difficult to distinguish in the 2BCCL product because of the ground clutter contamination in the radar signal (Sassen and Wang, 2008) as shown by Huang et al. (2015). In 2BCCL, the cloud base and top are given for up to ten cloudy layers, which is why we re-project these cloudy layers onto a 480 m vertical grid from 0 to 19.2 km to be consistent with the GOCCP and RL-GeoProf datasets described in sections 2.1 and 2.2. Those are then accumulated into a 2.5˚x2.5˚ grid as for the two other products. Note that there are substantial differences between results using the R04 and R05 versions, which is why the reader should refer to the original manuscript published in May, 22$^{nd}$ 2019 on the ESSD discussion website for results using the older R04 version.

## 3 Description of The Cumulus And Stratocumulus CloudSat-CAlipso Dataset (CASCCAD) discrimination algorithm

3.1 Why choose GOCCP and CloudSat-CALIPSO RL-GeoProf?

The main goal of this study is to document spatial distributions and profiles of Sc and Cu clouds on a global scale, with the desire to further analyze long-term relationships between Sc-Cu clouds and environmental parameters in future studies. For this purpose, we need to i) distinguish the two cloud types based on observable cloud-properties and ii) use datasets that are available for a time-period sufficiently long (~ 10 years) to compute statistically significant relationships (e.g., using 4 years of GOCCP rather than 10 may decrease the amplitude of the relationship between low clouds and SST anomalies by more than 15 %, Cesana et al., 2019a). Although both Sc and Cu clouds form within the planetary boundary layer (PBL), they have relatively different shapes as they are controlled by different physical mechanisms. However, one cannot separate clouds according to the mechanisms that form them as GCMs do using different PBL and convective parameterizations, which is why we choose to use the morphology to discriminate cloud types in this study. The Cu (Fig. 1, second to last column) can stretch up past the PBL into the lower free troposphere (z ~ 3 km) while they have typically a small horizontal extent (no more than a few km, e.g, Lamer et al., 2015; Nuijens et al., 2015b). On the contrary, the Sc (Fig. 1, first column) have a relatively small vertical extent (no more than a few hundred meters) and a cloud-top height (CTH) controlled by the PBL depth but spread out over tens to hundreds of kilometers (Wood, 2012) either homogeneously or heterogeneously (open-cell). In between, these two distinct regimes, various transitioning clouds may form (Albrecht et al., 2019; Teixeira et al., 2011) and the most frequent are: broken Sc and transition Sc-Cu, which is composed of Cu under Sc (Albrecht et al., 2019, Rauber et al., 2007) and Cu with stratiform outflow (Lamer et al., 2015, Nuijens et al., 2015b) (Fig. 1, second to fourth column, respectively). Clouds with a cloud base and a cloud top within and outside the lower free troposphere, respectively, are classified as deep Cu (Fig. 1, last column). Bearing the above facts in mind, we design the CASCCAD DA based on cloud height, vertical and horizontal cloud fraction and horizontal continuity, which can be applied to both GOCCP instantaneous profiles and the CloudSat-CALIPSO level 2 geometrical profile product (referred to as RL-GeoProf).

GOCCP instantaneous profiles satisfy the criteria i) and ii) mentioned above. They use all 70-m-large lidar shots every 333 m along-track without horizontal averaging, which allows the detection of the geometrically sparsest shallow Cu, besides the more horizontally extended Sc, over a relatively extended time-period (from June 2006 to 2017, while CALIPSO is still operating as of April 2019). This decadal dataset makes it possible to analyze climatological values of Cu and Sc cloud fraction and their relationships to environmental parameters. However, as the lidar penetrates within cloudy layers, the signal eventually attenuates completely for optical thickness greater than 3 to 5. Therefore, it is not always possible to observe the full troposphere with a space-borne lidar, which may cause differences in satellite-based cloud climatologies obtained from different instruments (Kikuchi et al., 2017; Thorsen et al., 2013). In these instances –i.e., in deep convective clouds or in the storm tracks–, the CPR capability complements cloud profiles beneath the height at which the lidar attenuates, although the CPR clutter prevents using CloudSat data below ~ 1000 m. Unfortunately, the RL-GeoProf product is only available for a short period of time (~ 4.5 years) due to the severe anomaly of April 2011, which is why CloudSat-CALIPSO observations satisfy i) but only partially ii). Note that the release R05 of RL-GeoProf came out after the submission of the original manuscript (late May 2019), which includes data after April 2011. However, these are for daytime only and substantial periods of time are still missing (e.g., May 2011 through May 2012 and the whole year 2014), which makes it difficult to compute a consistent climatology and derive statistical relationships between clouds and environmental variables. Additionally, the differences between CASCCAD using RL-GeoProf R04 and R05 are very small in the case studies of section 4.1 (see Fig. S2-S3-S4-S5). There is a small decrease of the overall cloud fraction (76.3 vs. 75.8 %, 37.2 vs. 35.2 %, 15.9 vs. 16 % and 80.8 vs. 77.2 % for R04 and R05 respectively), which affects mostly the Cu cloud fraction. Since the change is almost negligible, we decided not to update the global statistics with the R05 version for the sake of computational, space and time resources.

### 3.2 First criterion of the discrimination algorithm: the cloud-top height

As mentioned above, four main criteria –represented by different colors in Fig. 2– are used in the DA to separate Sc from Cu clouds and to characterize the various transitioning clouds presented in Fig. 2: the CTH, the Horizontal Cloud Fraction (HCF), the Vertical profiles of Cloud Fraction (VCF) and the horizontal continuity test. Note that the sensitivity to these criteria is later tested in section 4.1. The first step of the DA depends on the height of the cloud top (Fig. 2, 1[st] column, in grey). Because trade Cu and Sc are low clouds, their CTH must be within the lower free troposphere, which is defined as 3.36 km in GOCCP (approximately equivalent to the 680-hPa definition of Rossow and Schiffer, 1999). Furthermore, since Sc clouds cap the PBL, their CTH is typically lower than the PBL height over the main Sc deck areas, i.e., ~ 2 km (Albrecht et al., 1995; Bretherton et al., 2010; Garay et al., 2008; Wood, 2012; Zhou et al., 2015; Zuidema et al., 2009). Therefore, all cloud layers –i.e., a vertically-contiguous group of cloudy 480-m-pixels– with a cloud top higher than 1.92 km, which is the closest 480-m GOCCP level to 2 km, are diagnosed as Cu type. We remind the reader that the sensitivity of the algorithm to this criterion, as well as the other criteria, is tested in section 4.1.

## 3.3 Second criterion: the horizontal cloud fraction

The remaining clouds are passed to the second step of the DA (Fig. 2, 2nd column, in orange), which computes the HCF either centered around the lidar profiles (CHCF) or using forward (FHCF) or backward profiles (BHCF) as shown in Fig. 3. These three HCFs ensure capturing the full horizontal extent of the cloud layer regardless of whether the lidar probes toward the edge or the center of the layer while remaining more computationally efficient than treating clouds by contiguous horizontal-clusters. For example, the CHCF is 100 % in the specific case of Fig. 3a whereas FHCF and BHCF are about 50 %. Should the first lidar profile be at the beginning of the cloud, the centered, forward and backward HCF would be about 53, 100 and 7 % (Fig. 3b). Additionally, the HCFs are computed over four different length scales, 10, 20, 40 and 80 km, to characterize various cloud scenarios: open-cell and closed-cell Sc (Wood, 2012), and different Cu organizations in Sc-Cu transition clouds (Albrecht et al., 2019; Lamer et al., 2015; Rauber et al., 2007; Teixeira et al., 2011). The larger 40-80-km scales permit a clear distinction between the two type of clouds since Sc clouds typically cover vast areas compared to more fractionated trade-Cu clouds. Figures 4c and 4d show the probability density function (PDF) of the 40-km and 80-km CHCFs, respectively, for typical Cu (light blue bars) and Sc (light red bars) cases extracted from day and night CALIPSO orbits over the tropics (35°S/N, eight orbit segments in total). These results confirm a rather clear separation, marked by purple lines, between the two populations for CHCFs. Although some slight overlap is visible, it disappears when the 40 and 80-km CHCFs are run together (not shown). However, in regions of open-cell Sc and Sc-Cu transition, the overlap may be larger (Fig. S6) and additional tests are needed to determine the type of clouds, i.e., Sc, broken Sc, Cu under Sc, Cu with stratiform outflow or Cu. In those instances, the 10 and 20-km CHCFs help further distinguish Sc from the other clouds (Fig. 4a and 4b). Finally, note that when all 480-m-pixels below 1.92 km are fully attenuated (4 pixels), the profile is excluded from the HCF computation –consistent with what is done in the GOCCP product for cloud fraction computations– because we do not know whether these pixels are clear or cloudy.

## 3.4 Third criterion: the vertical profiles of cloud fraction

The third step of the DA (which comes into play only if the first two criteria have ruled out a pure Cu cloud) utilizes the VCF to capture the Cu under Sc and broken Sc clouds (Fig. 2, 3rd column, in purple). In our study, the Vertical profile of Cloud Fraction (VCF, sometimes referred to as vertical distribution of cloud fraction or cloud fraction profiles in previous literature) corresponds to the cloud fraction computed over 80-km along track at a particular level, in other words, it represents how often a cloud is encountered over 80 km in the horizontal direction at a particular level. Since Sc clouds are relatively shallow – no more than a few hundred meters (Wood, 2012) – any cloud layer with a substantial VCF over 3 levels or more (vertical extension greater than 1.44 km) is corrected from Sc to Cu under Sc while the rest remain diagnosed as Sc (Fig. 5a). The VCF threshold (= 0.12) is defined as approximately two times the standard deviation of the PDF of the 40-km CHCF computed using each of the seven first levels (0 to 3.36 km) in typical Cu regions. Note that because the CloudSat-CALIPSO vertical cloud fraction is less affected by attenuation, we defined a larger VCF threshold (= 0.3) for the RL-GeoProf CASCCAD

version. Finally, this VCF test is also applied to any cloud layer that passes the 40 and 80-km HCFs thresholds, regardless of their 10 and 20-km HCFs. In these cases, the cloud type is diagnosed as Cu under Sc if the VCF threshold is met, otherwise the cloud layer is diagnosed as broken Sc.

## 3.5 Fourth criterion: the horizontal continuity test

Once the cloud type is determined using the first three criteria, it is applied vertically to the whole cloudy layer. While the DA takes into account the horizontal extent of a cloud system via the computation of HCFs (including possible clear sky profiles), it does not track horizontally-contiguous clusters of clouds (without clear sky profiles). As a result, in Cu and transition Sc-Cu cases, the same horizontally-contiguous cloud layer may be diagnosed as both Sc-type (Sc and/or broken Sc) and Cu (Fig. 5b), which is more likely a Cu with a stratiform outflow (Lamer et al., 2015). To capture these particular clouds, we apply a horizontal continuity test (Fig. 2, 4th column, in green), which first detects a horizontally-contiguous cluster of clouds and then turns the non-Cu part (i.e., Sc, broken Sc or Cu under Sc) into a Cu with stratiform outflow if one third of the cluster is diagnosed as Cu type. We chose this arbitrary threshold because, on average, the fraction of the Cu that expands further aloft (geometrically thicker) is typically smaller than that near the lifting condensation level (Nuijens et al., 2015a) or that detrained near the trade-wind inversion (Nuijens et al., 2015a, Lamer et al., 2015).

## 4 Results

### 4.1 Case studies

To assess our CASCCAD DA, we analyze a series of three typical case studies: trade cumulus, stratocumulus and stratocumulus-cumulus transitioning clouds. First, we investigate the sensitivity of the DA to some of the criteria presented in Section 3 using GOCCP observations: the HCF (more or less conservative), CTH (one level higher) and VCF (smaller threshold, divided by 2) thresholds and the horizontal continuity test (turned off). We then compare the results of the standard DA applied separately to GOCCP and RL-GeoProf against the 2BCCL cloud types –for the same case studies– and utilize the collocated MODIS reflectance to provide a broader context of the cloud scene.

### 4.1.1 Stratocumulus case

Figure 6 shows the sensitivity of the GOCCP Sc-Cu Mask (Fig. 6b) to the different parameters used in the DA (Fig. 6c-d-e-f-g) for a daytime orbit segment off the coast of California during summer (see Fig. S7 for the exact location). The refined CALIPSO Science-Team (CALIPSO-ST) cloud mask ($\Delta z$ = 30 m; Vaughan et al., 2009) helps us get a better sense of the cloud geometrical thickness where the lidar is not fully attenuated (Fig. 6a, black color). Additionally, the MODIS true reflectance image confirms the presence of stratiform layers of clouds throughout the CALIPSO-CloudSat path (Fig. 7d). Except for the lower VCF parameters, which reduce the along-track Sc CF ($HCF_{Sc}$) by 5.6 % (absolute value, Fig. 6c), the $HCF_{Sc}$ is quite insensitive to the DA parameters ($HCF_{Sc}$ = 69.4 +0.3/-1.3 %). Reducing the VCF (Fig. 6c) turns the edges of

the Sc decks into transitioning Sc - Cu clouds (around 16˚N and 20˚N). However, changing the HCF thresholds have limited effect on the $HCF_{Sc}$ in this particular case (Fig. 6f and 6g).

GOCCP and RL-GeoProf have a similar $HCF_{Sc}$ while that of the 2BCCL product is smaller along with its total HCF (Fig. 7). The substantial difference between RL-GeoProf and 2BCCL total HCFs is mostly due to differences in lidar cloud fraction treatment. The lidar cloud fraction from RL-GeoProf comes from the CALIPSO 5km VFM mask, whereas that of 2BCCL comes from the Lidar-AUX product (Wang et al., 2013).

### 4.1.2 Cumulus case

As for the Sc case, the along-track Cu CF ($HCF_{Cu}$) is weakly sensitive to variations of the different DA parameters (Fig. 8, $HCF_{Cu}$ = 23.6 +1.1/-1.5 %). The most sensitive parameter is the horizontal continuity (Fig. 8e). When activated, it converts most of the large Cu between 20˚S and 18˚S from Sc to Cu with stratiform outflow although its southernmost edge remains likely incorrectly diagnosed as Sc. Most of the other cloudy features are diagnosed as Cu except for the cloudy layer located between 31˚ and 29˚S, which seems to be stratiform judging from its geometrical thickness (Fig. 8a) and reflectance (Fig. 9d). Here again, both GOCCP and RL-GeoProf diagnose similar Sc and Cu HCFs although the DA fully captures the aforementioned large Cu only when it is used with RL-GeoProf observations (Fig. 9b). Unlike the Sc case, GOCCP and RL-GeoProf disagree substantially with 2BCCL. For example, several cloud clusters are diagnosed as Sc by the 2BCCL algorithm although their geometrical thickness is larger than 1.5 km (Fig. 9c, around 34˚S, 19˚S and 12˚S), making it very unlikely that these clusters are actual Sc. As a result, the $HCF_{Sc}$ (17.2 %) is as large as the $HCF_{Cu}$ (18.3 %) and approximately three times larger than that of GOCCP and RL-GeoProf. An additional Cu case, in the trade-dominated NW Atlantic, confirms the ability of the DA to correctly diagnose a field of purely Cu clouds with no Sc (Fig. 10a and 10b). Contrary to Fig. 9c, the 2BCCL product does not detect Sc-type clouds (Fig. 10c, $HCF_{Sc}$ = 0 %) and even underestimates the Cu cloud fraction ($HCF_{Cu}$ = 10 %) compared to GOCCP and RL-GeoProf ($HCF_{Cu}$ = 19.5 % and 15.9 %, respectively). This case also highlights one of two notable differences between the GOCCP and RL-GeoProf CASSCAD datasets: the latter misses a significant amount of cloud below 1 km altitude in scattered cumulus environments, presumably due to a combination of the radar surface clutter limitation and the lidar $CF_{lidar}$ threshold used to diagnose a pixel as cloudy ($CF_{lidar}$ = 0.5, cf section 2.2).

### 4.1.3 Cumulus and open-cell stratocumulus case

The last case study extends from the subtropics to the extra-tropics. Such location allows us to characterize transitioning Sc-Cu cases, which includes Sc, open-Sc and the different Cu-type clouds (Fig. 11). A visual inspection of the CTH variation from the CALIPSO VFM (Fig. 11a) suggests that this orbit segment contains three distinct clusters of clouds: Sc from 55˚ to 43˚S and from 25˚ to 20˚S and Cu in between. These three distinct layers are quite well captured by the DA although the DA is more sensitive to changes in the parameters than in the other cases. The most sensitive parameters are the VCF and HCF thresholds. Reducing the VCF threshold (Fig. 11c) turns 7.6 % of the Sc into Cu (absolute value) mostly poleward of 43˚S because the BL height decreases, causing multiple levels to be cloudy and subsequently diagnosed as Cu. Choosing smaller

HCF thresholds (Fig. 11g) increases the Sc amount by 5.2 % (absolute value). This converts the few Cu poleward 43˚S into Sc as well as some Cu around 24˚S, which could very well be "true" Sc.

The three clusters of clouds are also well captured in the RL-GeoProf dataset, which detects somewhat a little more Cu and Sc than GOCCP making the total HCF larger as well (Fig. 12), possibly due to the $CF_{lidar}$ threshold, which may cause an overestimation of the cloud fraction. Also note that the additional Cu-type clouds detected by RL-GeoProf product are mostly classified as Cu with stratiform outflow. On the contrary, the 2BCCL product diagnoses nearly two times more Sc than the CASCCAD datasets and two to three times less Cu while its total HCF is in between the two datasets.

## 4.2 Statistical analysis

### 4.2.1 Maps

In this section, we analyze climatological geographical distributions of Sc and Cu clouds for the three products presented before as well as for a subset of passive-sensor observations. For the CASCCAD datasets, we treat transitioning clouds as distinct cloud types and thus do not include them in the Sc and Cu cloud fractions to be discussed below, although one could easily include the broken Sc in the Sc cloud fraction and the Cu under Sc as well as the Cu with stratiform outflow in the Cu cloud fraction. The passive-sensor observations include the International Satellite Cloud Climatology Project (ISCCP, Rossow and Schiffer, 1999), the Moderate Resolution Imaging Spectroradiometer (MODIS, King et al., 2013) and the Multi-Angle Imaging Spectroradiometer (MISR, Marchand et al., 2010) observations. In the passive remote sensing datasets, Sc and Cu are separated using a cloud top pressure (CTP) - cloud optical thickness (COT) diagram introduced by Rossow and Schiffer (1999): CTP must be larger than 680hPa for each type and COT smaller or larger than 3.6 for Cu and Sc clouds, respectively. There are advantages and disadvantages to each approach. In general, passive remote-sensing instruments provide better temporal and spatial sampling than active sensors and the data products exist for a longer period of time. On the other hand, passive sensors can have difficulty identifying cloud top altitude, especially in multi-layer situations, relative to active sensors. For our purposes, we note that the COT-based method has been shown to mis-classify Cu and Sc that have moderate optical thickness (e.g., Pincus et al., 1999). In addition, Mace and Wrenn (2013) showed that except for thin cirrus and Sc cloud types, the COT-derived cloud types are mostly mixtures of different cloud types in two regions of the Eastern Pacific. However, passive-sensor estimates of Sc and Cu provide a broader context and help us emphasize the added value of new Sc-Cu discrimination methods based on active-sensor satellites.

Overall, all products identify quite well the large cloud fraction in the tropical and subtropical stratocumulus areas, off the west coast of the continents (Fig. 13, top row). RL-GeoProf and 2BCCL products detect the largest low-level cloud fraction (Fig. 14, zonal and global mean, top row) although they might overestimate the fractionated clouds (e.g., Cu). Unlike the passive-sensor products –based solely on the CTP and COT– the active-sensor products do not detect a significant amount of Cu off the west coasts of the continents compared to the large Sc cloud fraction, which ranges from 50 % off the coast of Australia up to 85 % in the heart of the deck off the coast of Peru in both GOCCP and RL-GeoProf (Fig. 13, third row). These results are somewhat different from a previous analysis in which the Sc cloud fraction ranges from 40 to 60 % over the Sc

deck areas (Wood, 2012). On the contrary, the CASCCAD products place a substantial amount of Cu clouds west of the Sc decks (up to 40 %) and in the trade-wind regions (between 20 and 30 %), similar to MISR observations (Fig. 13, third row), which is more sensitive to fractionated clouds than ISCCP and MODIS, while 2BCCL only classifies a small amount of Cu clouds (between 10 and 15 %) in these regions. Here again our findings somewhat contradict earlier results retrieving Sc clouds

20 % of the time in the trade-wind regions (Wood, 2012). Besides the Cu and Sc categories, the CASCCAD products have a third category referred to as transitional clouds (Fig. 13, fourth row), which is supposed to capture regions of transition between Cu and Sc clouds. As expected, these clouds are located between Sc decks and trade-wind regions and in the extra-tropics, where one could expect the two types of clouds to co-exist. However, they represent a small part of the total low-cloud fraction in the tropics while it is larger in the extra-tropics (between 10 and 20 %, Fig. 14, fourth row), where both Sc and Cu types

have a similar and substantial cloud fraction (15 to 30 %). As mentioned in the cumulus and open-cell stratocumulus case, RL-GeoProf CASCCAD classifies more transitioning clouds than its GOCCP counterpart. This is mainly due to the fact that RL-GeoProf contains larger cloud clusters than GOCCP, making the continuity test more efficient (e.g., the large Cu cloud cluster around 19˚S in Fig. 9 and the Cu part of the open-cell and Cu case, Fig. 12) and also because RL-GeoProf is less affected by overlapping mid- and high-cloud attenuation and better capture the full vertical extent of low clouds. Finally, the ratio of Sc

clouds to Sc and Cu clouds (transitioning clouds being excluded) document the regions dominated by each type of clouds (Fig. 13, bottom row). Such information could be very useful to evaluate GCMs, which struggle to reproduce the Sc-Cu transition (e.g., Teixeira et al., 2011). The CASCCAD products robustly describe the tropical open ocean regions being almost exclusively dominated by Cu clouds while the oceans off the west coasts of the continents are mostly covered by Sc clouds. Such a picture is consistent with previous results from field campaigns, e.g., along the Global Energy and Water Experiment

(GEWEX) Cloud System Studies (GCSS) Pacific Cross-Section Intercomparison (GPCI) transect (e.g., Zhou et al., 2015), or ship-based observations in the southeastern Pacific (Garay et al., 2008) and ground-based data, e.g., over Barbados (e.g., Nuijens et al., 2015). Finally, the CASCCAD products classify less Sc than the other products (Fig. 13 and 14, second row) in the extra-tropics and polar regions (poleward of 35˚).

In the 2BCCL, MODIS and ISCCP products, most of the globe is dominated by Sc clouds, as opposed to the CASCCAD

products and MISR, which depict the tropics dominated by Cu clouds and the extra-tropics by Sc clouds. The passive-sensor results suggest that the optical thickness does not permit a clear distinction between Cu and Sc clouds, in agreement with earlier studies (e.g., Pincus et al., 1999), although clustering analysis from passive-sensor observations may represent a better alternative (Tselioudis et al., 2013, their Fig. 3). To further investigate this statement, we classify GOCCP low clouds as a function of their opacity. The low-cloud containing profiles that are diagnosed as opaque (i.e., no surface echo retrieved) have

an optical thickness larger than approximately 3 (Guzman et al., 2017). This optical thickness is further used as a threshold to separate Sc (opaque clouds with COT > 3) from Cu clouds (thin clouds with COT < 3), which is about the same optical thickness used in the passive sensor (i.e, COT = 3.6) to distinguish Cu and Sc clouds (i.e., COT = 3.6). As for the passive-sensor satellite observations, this method does not allow a clear separation between the two cloud populations (Fig. S8) although the ratio of Sc to Sc and Cu of derived from the two methods are well-correlated (~ 0.65). However, it confirms that

trade-wind regions have smaller opacity and therefore have a different radiative impact on surface and TOA fluxes than more opaque Sc-dominated regions.

The newer clustering approaches, such as the ISCCP weather states (ISCCP-WS, Tselioudis et al., 2013) and MODIS cloud regimes (MODIS-CR, Oreopoulos et al., 2014), do not discriminate cloud types or even low cloud from middle and high clouds. Instead, they represent mixtures of cloud types although one cloud type is often prevalent in a given cluster. As such, it is not straightforward to directly compare them with the CASCCAD datasets. Yet, we believe that showing the results from the CR and WS approaches provides a broader context and additional information. For this reason, we analyze the ISCCP WS and MODIS CR observations against CASCCAD datasets in Fig. 15. The ISCCP-WS cloud fractions are obtained by multiplying the monthly relative frequency of occurrence (RFO) of WS7-8 for Cu and WS9-10-11 for Sc by the monthly total cloud fraction, based on Tselioudis et al. (2013). A similar method is applied for MODIS-CR cloud fractions with CR11 for Cu and CR7-8-9-10 for Sc, based on Oreopoulos et al. (2014). The clustering-derived datasets show an overall better agreement with CALIPSO and CloudSat-CALIPSO CASCCAD, particularly for the Sc regimes (Fig. 14 and Fig. 15). In the tropics, the Cu regimes are better correlated with CALIPSO CASCCAD in terms of geographical distribution (from r = 0.26 to 0.63 for ISCCP and r = -0.02 to 0.35 for MODIS) although MODIS underestimates their fraction while ISCCP overestimates it (likely due to mid and high-level clouds contained in WS7 and WS8). Note that CR12 is currently investigated to be further decomposed in CR12a-b-c and some of these sub-CRs could correspond to Cu-type regimes (Oreopoulos et al., personal communication).

### 4.2.2 Profiles

Figure 16 shows global zonal profiles of cloud fraction for Cu, Sc, transitioning and all low-level clouds as observed by GOCCP, RL-GeoProf and 2BCCL, for the first time. As in Section 4.2.1, the transitioning clouds are not accounted for in the Sc and Cu cloud fractions of the CASCCAD datasets. Consistent with the map analysis, 2BCCL observations retrieve clouds in the low-levels more frequently than GOCCP (Fig. 16, top row) while the difference with RL-GeoProf is rather small compared to that with GOCCP (approximately two times larger). The large difference between GOCCP and RL-GeoProf cloud fractions in the low levels mostly comes from mid and high-level topped clouds (e.g., frontal clouds in the extra-tropics and cumulonimbus and congestus clouds in the tropics, Fig. S9), which typically obscure CALIPSO vision by attenuating the lidar beam before it reaches the low levels. When separated into cloud types, GOCCP and RL-GeoProf observations agree quite well for Sc and transitioning clouds globally (both in terms of pattern and amount) and for Cu clouds in the deep tropics (15˚S/N) and down to 2 km in the subtropics and the extra-tropics (Fig. 16, 2nd row). Between 2 km and 1 km, RL-GeoProf diagnoses more Cu than GOCCP. Such a difference is due to the different sensitivities of the lidar and radar instruments to clouds. The lidar signal becomes quickly attenuated by the optically and geometrically thick Cu –besides the attenuation from overlapping mid or high-clouds– whereas the CloudSat radar continues detecting clouds down to 1 km. Below 1 km the surface clutter and the lidar attenuation make it difficult to retrieve a reliable cloud fraction.

Consistent with the case study and geographical analysis (Section 4.2.1), the 2BCCL product diagnoses far more (less) Sc (Cu) than the CASCCAD products, making the ratio of Sc to Sc-and-Cu clouds largely dominated by Sc (Fig. 16, bottom row). Furthermore, the vertical distribution of Sc and Cu clouds substantially differs from that of the CASCCAD products. 2BCCL Sc clouds may extend up to 3 km while the most part of Cu clouds are concentrated around 1 km and almost exclusively in the tropics. This lack of Cu and excess of Sc clouds above 1 km in tropical subsidence regimes (Fig. 16, right column) is in disagreement with the CASCCAD products but also with previous studies focused on Sc regions (Cesana et al., 2019a, their Fig. 6) and Cu regions (Nuijens et al., 2015a, their Fig. 2, see also Fig. S1)(Nuijens et al., 2015a). Therefore, one may think that 2BCCL overestimates (underestimates) Sc (Cu) clouds by mis-diagnosing some Cu clouds into Sc clouds. On the contrary, the CASCCAD products better match typical profiles of Sc and Cu clouds.

Finally, these global scale profiles are consistent with the physical processes controlling each type of cloud and measurements from previous literature. The Sc clouds –driven by radiative cooling– cap the PBL, a little higher than 1 km in the tropics and lower toward the poles, and their geometrical thickness is smaller than 1 km. Averaged over the tropical subsidence regimes (and in the extra-tropics, Fig. S10), the Sc vertical cloud fraction peaks between 10 and 12 %, depending on the dataset, whereas it is larger for Sc deck areas only (not shown). On the contrary, Cu cloud base –forced by surface fluxes– mostly form below 1 km, near the LCL, and vertically extend further aloft (around 2.5 km). Although the Cu map cloud fraction is smaller than that of Sc in the tropics (Fig. 13, compare the second and third rows), their vertical cloud fraction is about the same, around 8 %, because they cover a larger domain (Fig. 16, right column). While the Sc vertical cloud fraction remains unchanged in the extra-tropics, its Cu and transitioning counterparts appear slightly larger, around 10 % and between 6 and 10% (Fig. S10), respectively.

**5 Conclusion**

In this paper, we document spatial distributions and profiles of stratocumulus (Sc), cumulus (Cu) and transitioning (i.e., broken Sc, Cu under Sc and Cu with stratiform outflow) clouds on a global scale. To this end, we design a discrimination algorithm (DA; Section 3) that distinguishes Sc and Cu based on four observable cloud-properties: cloud top height (CTH), horizontal cloud fraction (HCF), vertical cloud fraction variability (VCF) and horizontal continuity. These simple criteria are sufficient to characterize the distinctive shape of Cu, which have a limited horizontal extent and highly variable CTH as opposed to Sc, which cover larger areas and have a small and stable geometrical thickness. The DA is utilized separately on instantaneous profiles of active-sensor CALIPSO-GOCCP (Guzman et al., 2017) and CloudSat-CALIPSO combined observations (RL-GeoProf; Cesana et al., 2019b; Mace and Zhang, 2014) to create the distinct GOCCP and RL-GeoProf Cumulus And Stratocumulus CloudSat-CAlipso Datasets (CASCCAD) .

The choice of DA parameters is then investigated in Cu, Sc and Sc-Cu transition case studies (Section 4.1), supported by additional viewings from MODIS true reflectance and full resolution CALIPSO VFM ($\Delta z = 30m$). The results show that the

DA robustly captures Sc, Cu and Sc-Cu transition clouds although the choice of VCF and HCF thresholds may slightly affect the Sc-Cu partitioning in open-Sc and Cu regions.

The distinct GOCCP and RL-GeoProf CASCCAD global-scale statistics (Section 4.2) are then compared to a subset of passive-sensor satellite datasets and to the only existing CloudSat-CALIPSO cloud-type climatology 2B-CLDCLASS-LIDAR (2BCCL, Sassen and Wang, 2008). In passive-sensor satellite observations, Sc and Cu co-exist everywhere and no region is fully dominated by a particular type of cloud (Fig. 12, bottom row) when distinguished only based on their cloud optical thickness while Sc are mostly confined to specific regions (off the west coast of continents) with the more recent clustering approach. On the contrary, Sc clouds largely dominate the global statistics from the 2BCCL point of view, which may mis-diagnose a substantial portion of Cu clouds as Sc clouds in the trade-wind and extra-tropical regions. Interestingly, the CASCCAD observations depict tropical oceans being almost exclusively dominated by Cu clouds (around 20 % on average and up to 40 %) while the oceans off the west coasts of the continents are mostly covered by Sc clouds (50 to 85 %), with transitioning clouds in between (10 to 15 %). Our results provide a broader context to earlier findings from ground-based and field campaigns (Albrecht et al., 2019, 1995; Bretherton et al., 2010; Comstock et al., 2004; Garay et al., 2008; Wood, 2012; Zhou et al., 2015; Zuidema et al., 2009). For example, our globally-averaged profiles of Cu cloud fraction over the tropical oceans are almost identical to that found by Nuijens et al. (2015a) over the Barbados, in terms of shape (cloud base below 1 km and cloud top above 2 km) and frequency of occurrence (~ 10 %). Another interesting result concerns the distribution and magnitude of Sc cloud fraction. Our results indicate that the Sc clouds occur up to 85 % of the time over Sc deck areas compared to 60 % in earlier studies (i.e., Wood, 2012) and that their presence in trade-wind regions is negligible as opposed to a 20 % cloud frequency (i.e., Wood, 2012). Furthermore, our analysis indicates that the optical thickness, albeit useful, is not a sufficient parameter to discriminate Cu from Sc clouds, in agreement with previous literature (e.g., Pincus et al., 1999). Finally, one of the reasons we developed CASCCAD is to provide an improved observational constraint for low-level cloud feedbacks in GCMs. Although the CASCCAD DA cannot be implemented in a lidar or radar simulator (Chepfer et al., 2008; Marchand et al., 2008), it is still possible to use CASCCAD datasets for model evaluation because i) both the convective and stratiform cloud fraction are provided as inputs to the lidar simulator and could be easily saved separately rather than summed up; and ii) a simulator is not necessarily needed for model-to-observation comparison of Cu and Sc clouds over the tropical oceans, because we identify the different cloud modes explicitly and we can select regimes in which lidar attenuation is negligible (e.g., $\omega_{500} > 0$ hPa/day, Cesana et al., 2019a). Therefore, the two CASCCAD datasets make it possible to evaluate the shallow convection (Cu type) and boundary layer (Sc type) clouds in state-of-the art climate models, which are typically generated by distinct parametrizations (i.e., Cesana et al., 2019a), but also in unified-scheme turbulence models, in which Sc and Cu are still generated by distinct physical mechanisms (eddy diffusion for Sc and mass flux for Cu, e.g., Köhler et al., 2011). In such parameterizations, the existence of shallow convection is determined by a large-scale environmental index of inversion strength. In other unified turbulence schemes, the diagnosis of model success is achieved by defining Sc, Sc-Cu, and Cu regimes in terms of inversion strength (Bogenschutz and Krueger, 2013). We suggest that a dataset such as CASCCAD that directly identifies these physically different cloud types is a better metric to use to judge model realism and fidelity. By

doing so, one could also assess the radiative contribution of Sc and Cu clouds to climate and potentially improve our understanding of low-level cloud feedbacks.

**Data availability**

The distinct GOCCP and RL-GeoProf CASCCAD statistical datasets (Cesana, 2019) can be downloaded on the GISS website (https://data.giss.nasa.gov/clouds/casccad/) and on the zenodo website (https://zenodo.org/record/2667637; DOI: http://doi.org/10.5281/zenodo.2667637).

GOCCP instantaneous profiles used to produce the CASCCAD dataset were downloaded from the CFMIP-Obs website (http://climserv.ipsl.polytechnique.fr/cfmip-obs/Calipso_goccp.html). The CloudSat–CALIPSO data used to produce the

CASCCAD dataset (i.e., 2B-GEOPROF and 2B-GEOPROF-LIDAR) and the 2B-CLDCLASS-LIDAR product were obtained from the CloudSat Data Processing Center (http://www.cloudsat.cira.colostate.edu/data-products/level-2b).

The ISCCP WS data were obtained on the official ISCCP website (https://isccp.giss.nasa.gov/wstates/gcluster.html) and the matching total cloud fraction from the CFMIP-OBS website (https://climserv.ipsl.polytechnique.fr/cfmip-obs/Isccp.html). The MODIS CR and matching total cloud fraction were provided by Lazaros Oreopoulos and Nayeong Cho who can make them

available upon request by contacting them at Lazaros.Oreopoulos@nasa.gov.

**Author contributions**

GC designed the study, developed the products and carried out the analysis with inputs from AD and HC. GC wrote the manuscript with contributions from AD and HC.

**Competing interests**

The authors declare that they have no conflict of interest.

**Acknowledgements**

GC and AD were supported by a CloudSat-CALIPSO RTOP at the NASA Goddard Institute for Space Studies. The authors would like to thank NASA and CNES for giving access to CALIPSO and CloudSat observations, and Climserv for giving

access to CALIPSO-GOCCP observations and for providing computing resources. The authors also thank Lazaros Oreopoulos and George Tselioudis for their help in processing the MODIS CR and ISCCP WS datasets, respectively. GC thanks Zhien

Wang for his help in understanding the 2B-CLDCLASS-LIDAR dataset and Andy Ackerman, Ann Fridlind, Max Kelley and Greg Elsaesser for helpful comments and discussion.

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

**Figures**

**Figure 1: Cartoon representing the different cloud morphologies used in this study. The blue dashed line denotes the upper limit of the low-level clouds following GOCCP definition (3.36 km). Note that the Deep Cu type is not a low cloud, but either a mid- or high-level cloud.**

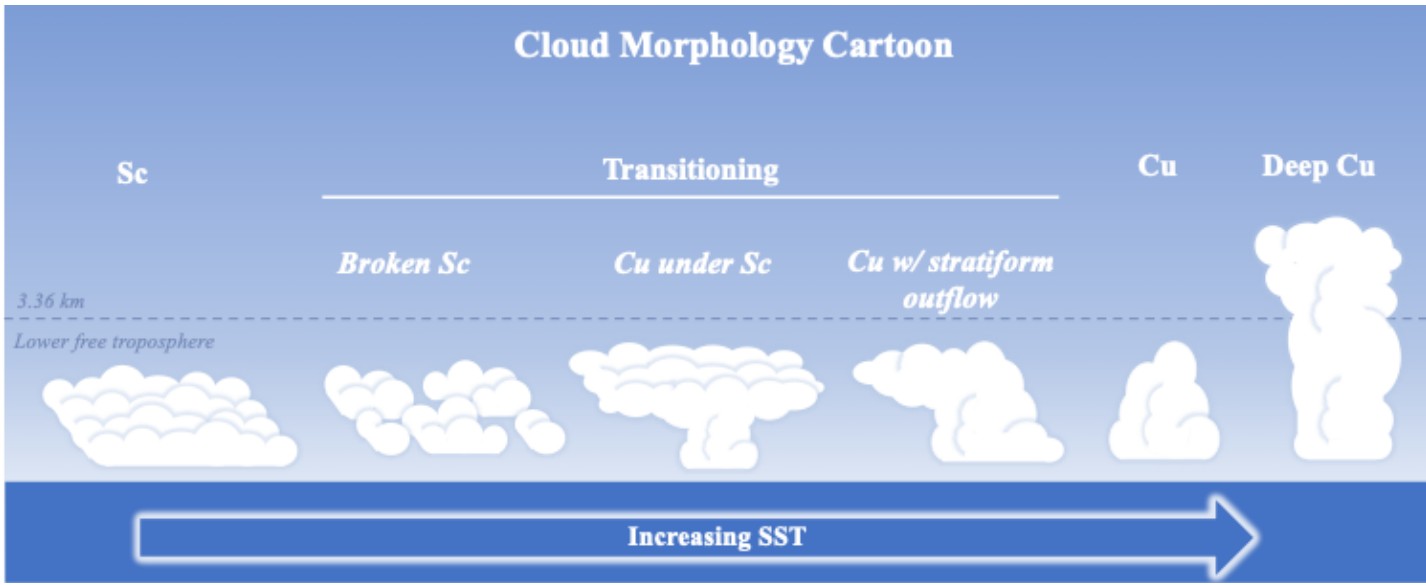

**Figure 2: Flow diagram of the algorithm that performs the Sc-Cu discrimination. The grey, orange, purple and green colors represent the different steps of the discrimination algorithm to separate Sc, broken Sc, Cu, Cu under Sc and Cu with stratiform outflow clouds: height, horizontal extent, vertical extent and horizontal continuity, respectively. The choice of HCF thresholds is discussed in section 3.3 and theirs values are shown in Fig. 4. The VCF threshold (0.12) and the horizontal continuity test are also explained in section 3.4 and further used in Fig. 5. Note that the steps 1 to 3 are applied to every profile while the step 4 is only applied to contiguous cloud clusters.**

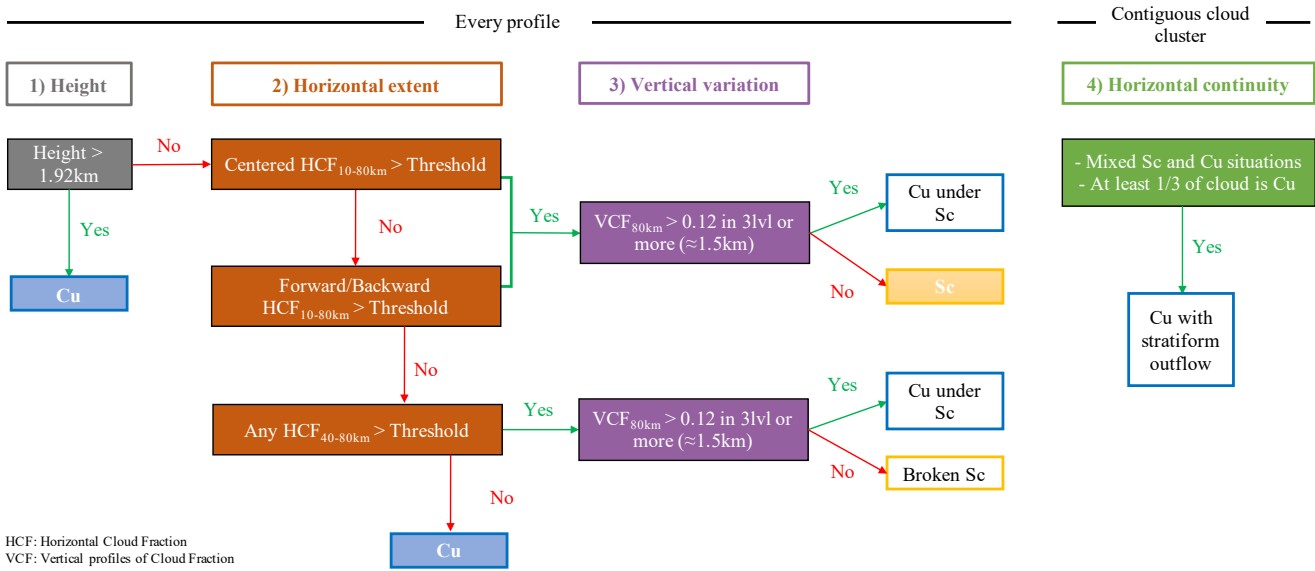

**Figure 3: Cartoon of the horizontal along-track averaging that is used by the DA to compute the horizontal cloud fraction (HCF): a) the lidar beam is in the middle of the cloud and b) the lidar beam is at the edge of the cloud. The dark-green circles correspond to the profiles used to in the different horizontal averaging scenarios (centered, forward or backward).**

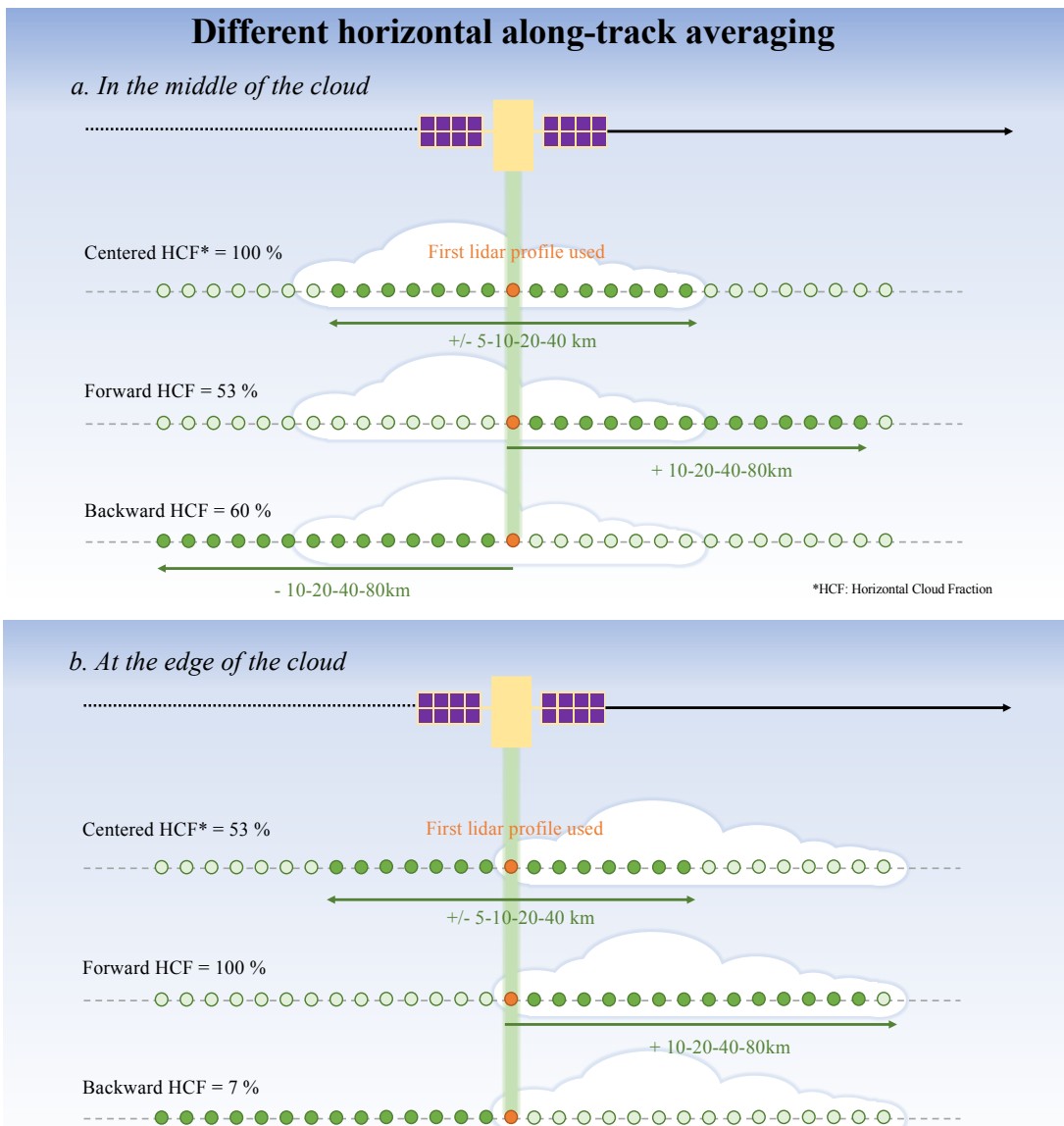

**Figure 4: PDF of the different along-track CHCFs for Cu (blue) and Sc (light red) typical regions computed from eight orbit segments. The thresholds are represented in purple, at approximately the overlap of the two distributions.**

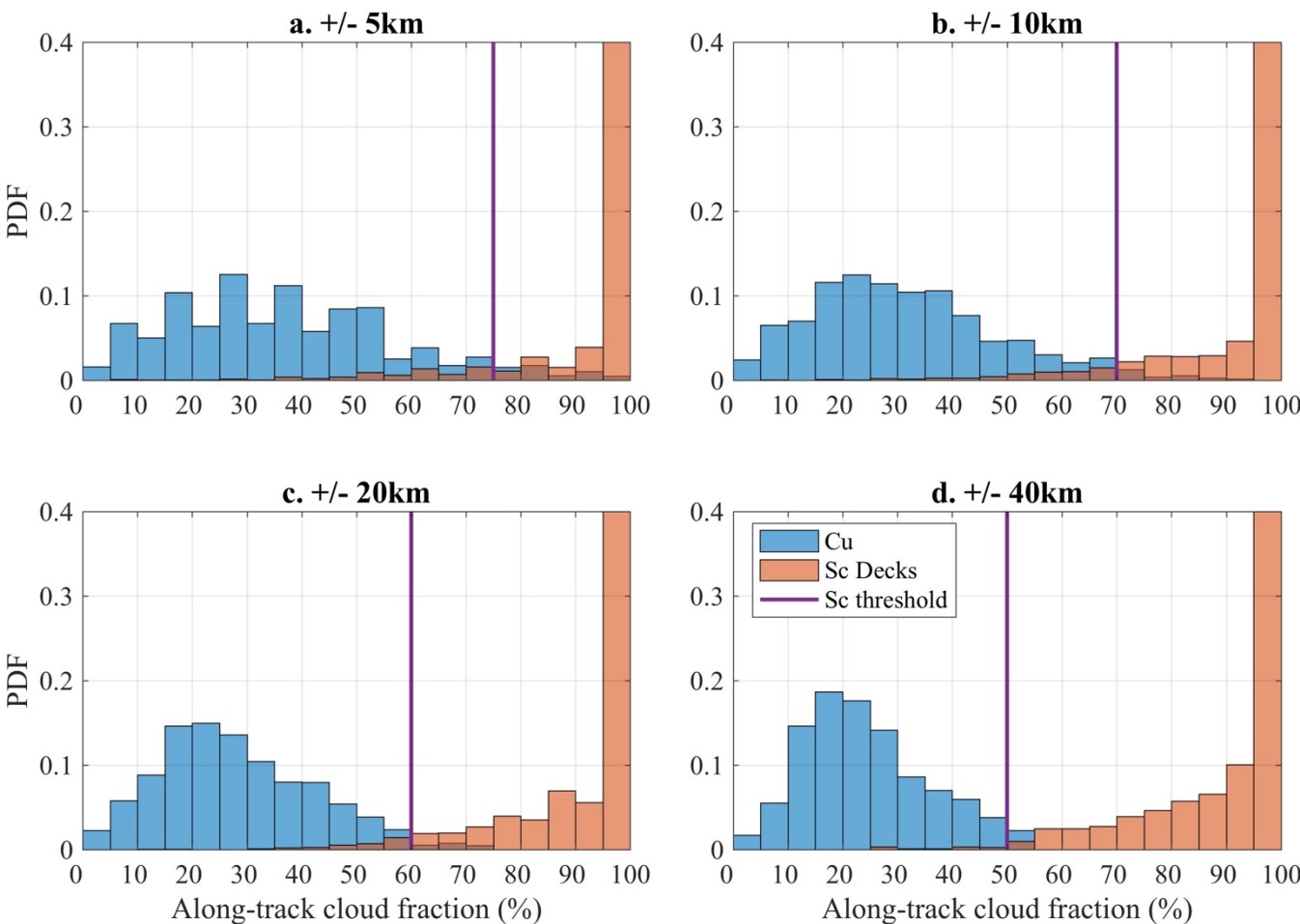

**Figure 5: Cartoon of the Cu and Sc mixed situations using additional diagnostics in the DA: a) Cu under Sc and b) Cu with stratiform outflow. After the DA diagnoses (1) an initial cloud type, the additional (2) "significant VCF variability" test and (3) "horizontal continuity" correction are applied to contiguous cloud clusters (See details in section 3.4 and 3.5).**

## a. Cu under Sc

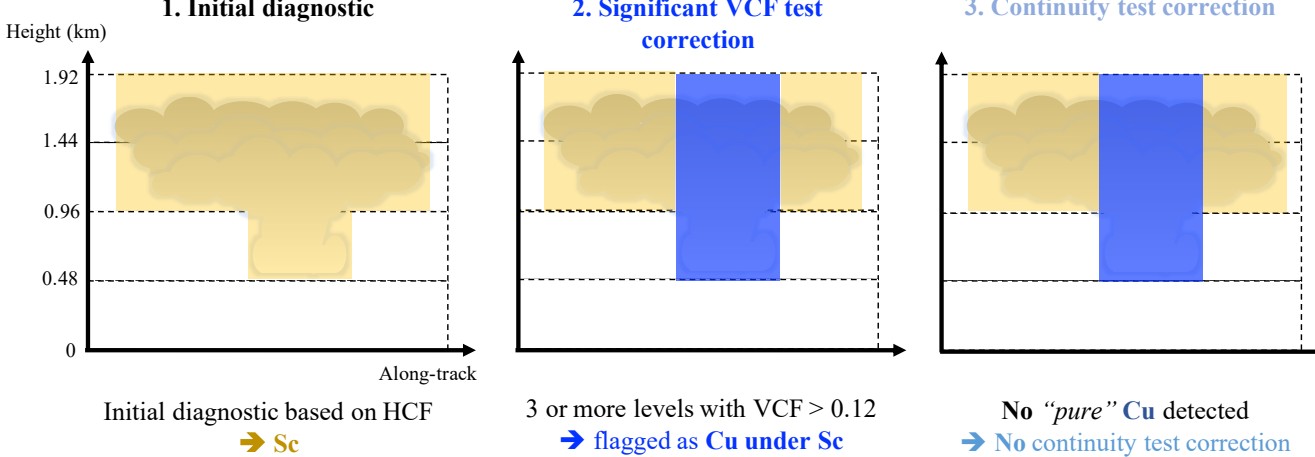

## b. Cu with stratiform outflow

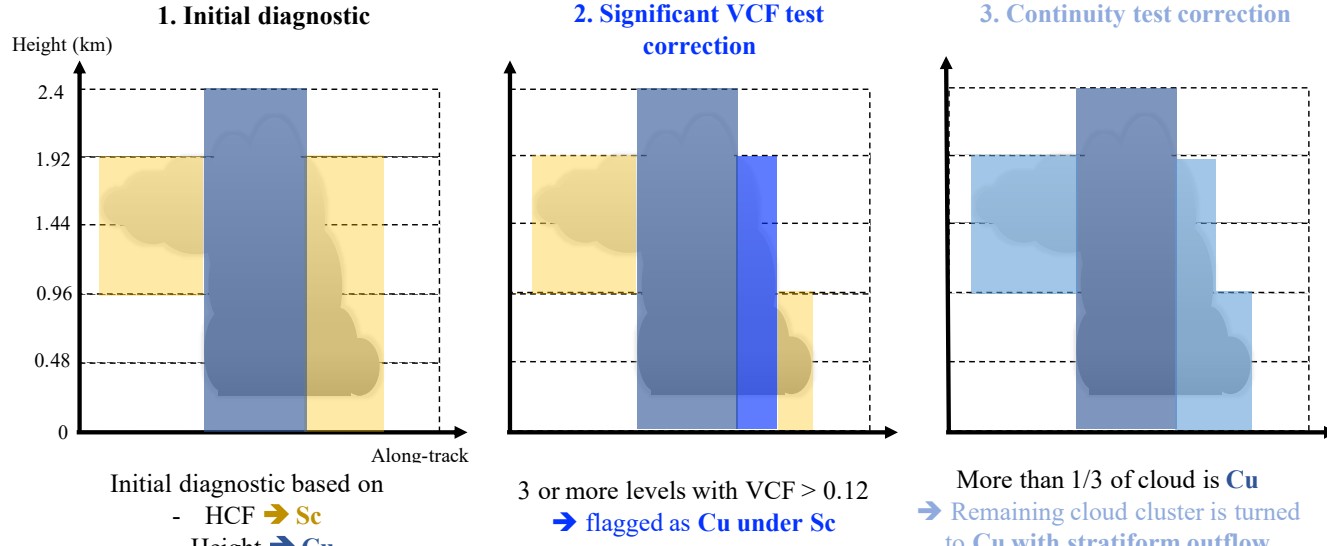

**Figure 6: Daytime orbit segment off the coast of California (~ 8˚N to 46˚N, 2008-07-08 21:47:30) showing a typical stratocumulus case. (a) VFM CALIPSO-ST at 30-m vertical resolution, (b) to (g) sensitivity of the GOCCP Sc-Cu mask to different DA parameters: (b) the VCF threshold is divided by 2, (c) the CTH threshold is elevated from 1.92 km to 2.4 km, (d) standard DA parameters, (e) the continuity test is turned off, (f) the 10-km and 20-km HCF thresholds are more conservative (i.e., increased by 0.1, absolute value) and (g) the 10-km and 20-km HCF thresholds are less conservative i.e., (decreased by 0.1, absolute value). The HCF of Sc-dominated (including broken Sc) and Cu-dominated (including Cu under Sc and Cu with stratiform outflow) clouds are given in each subplot (top left corner). Reddish and bluish pixels correspond to Sc-dominated and Cu-dominated type of clouds, respectively. From the top to the bottom, the VFM color bar's labels correspond to undetermined, aerosol, cloudy, clear and fully attenuated pixels. From the top to the bottom, the GOCCP Sc-Cu mask color bar's labels correspond to cumulus, Cu under Sc, Cu with stratiform outflow, broken Sc, stratocumulus, and clear pixels.**

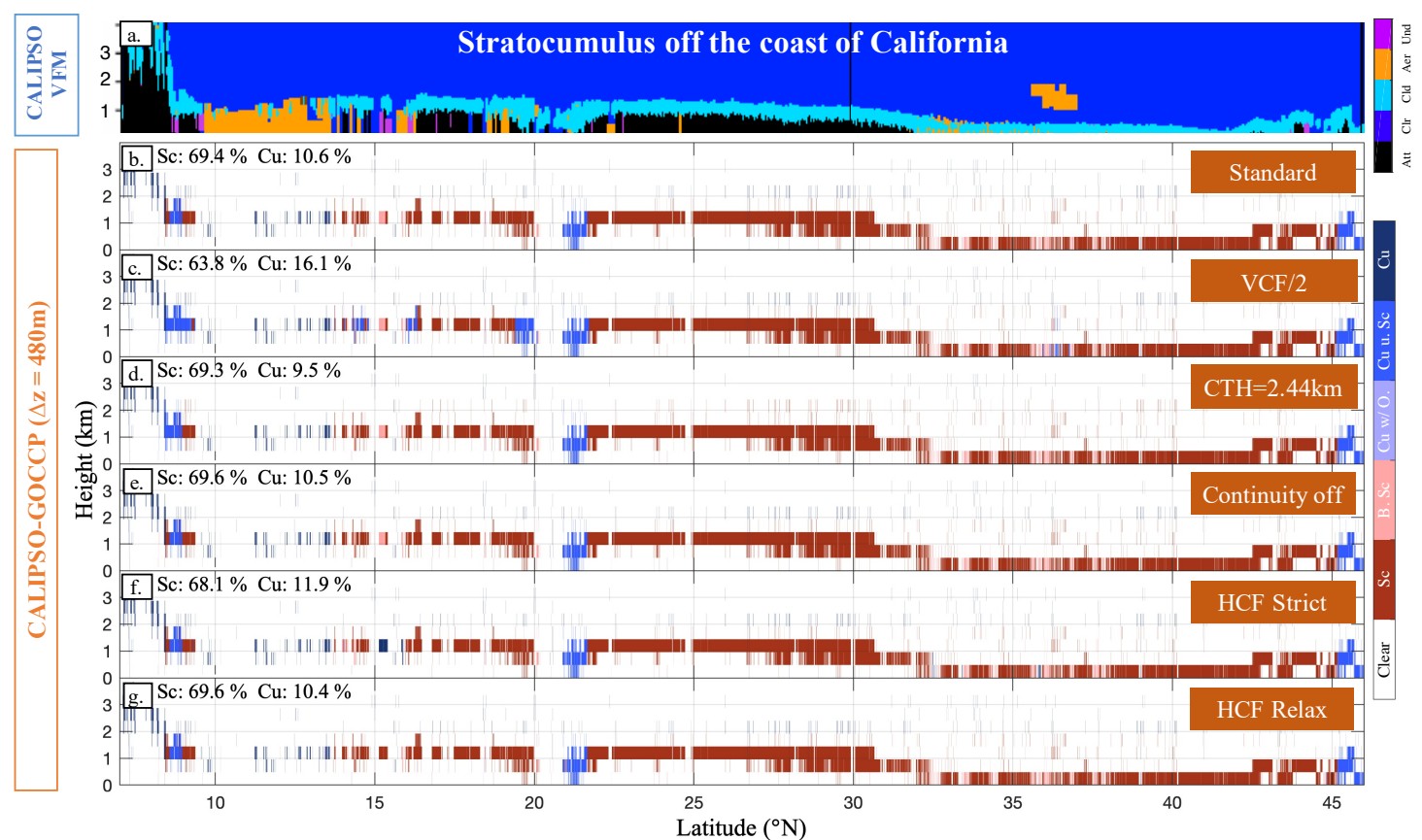

**Figure 7: Same stratocumulus case study as in Fig. 5 but for (a) the standard GOCCP Sc-Cu mask (same as Fig. 5b), (b) the RL-GeoProf Sc-Cu mask, (c) 2BCCL cloud type mask and (d) MODIS true reflectance. The red line corresponds to the CALIPSO-CloudSat overpass.**

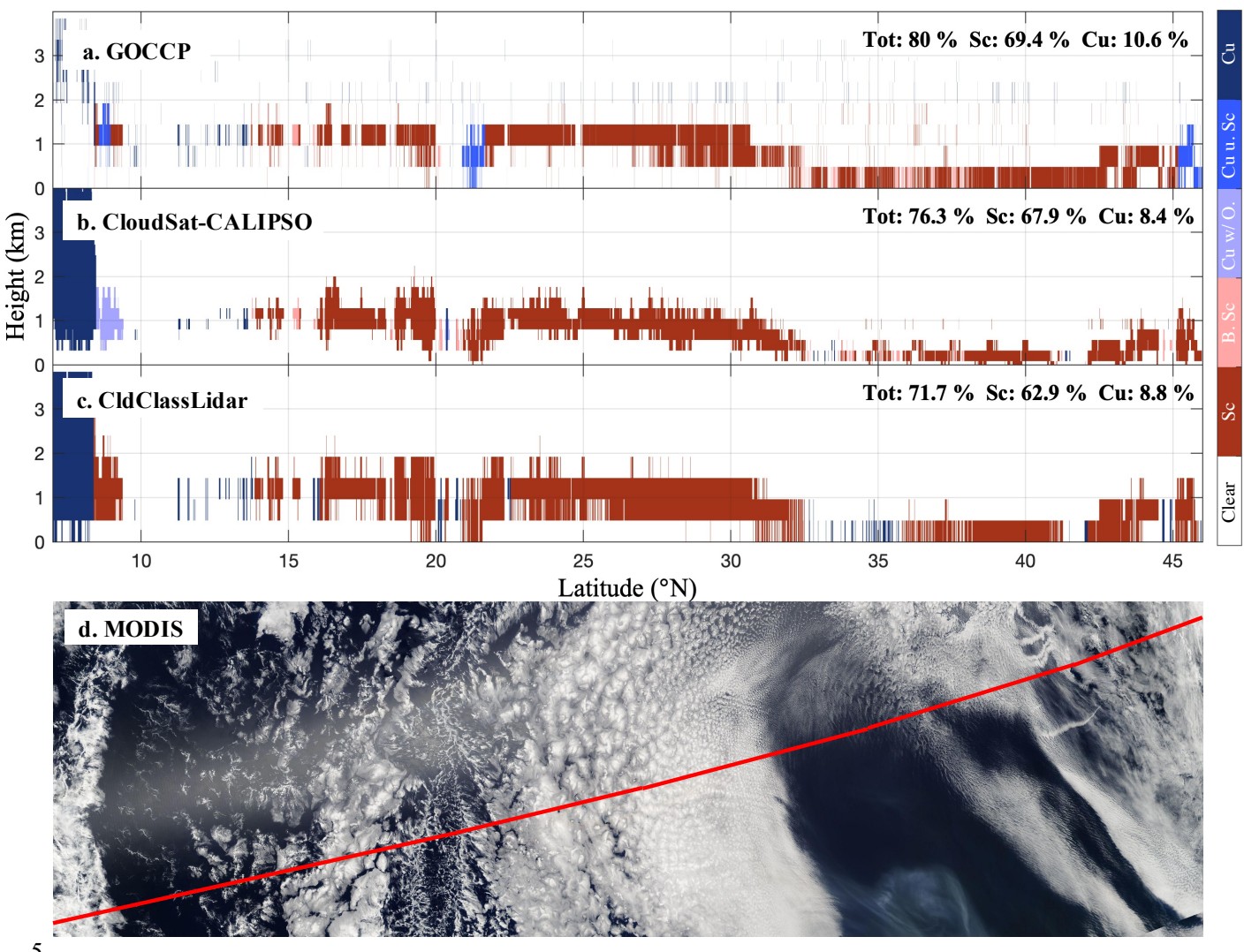

**Figure 8: Same as Fig. 5 but for a typical cumulus case in the south-east Pacific (~ 37°S to 8°S, 2008-07-08 21:47:30, daytime).**

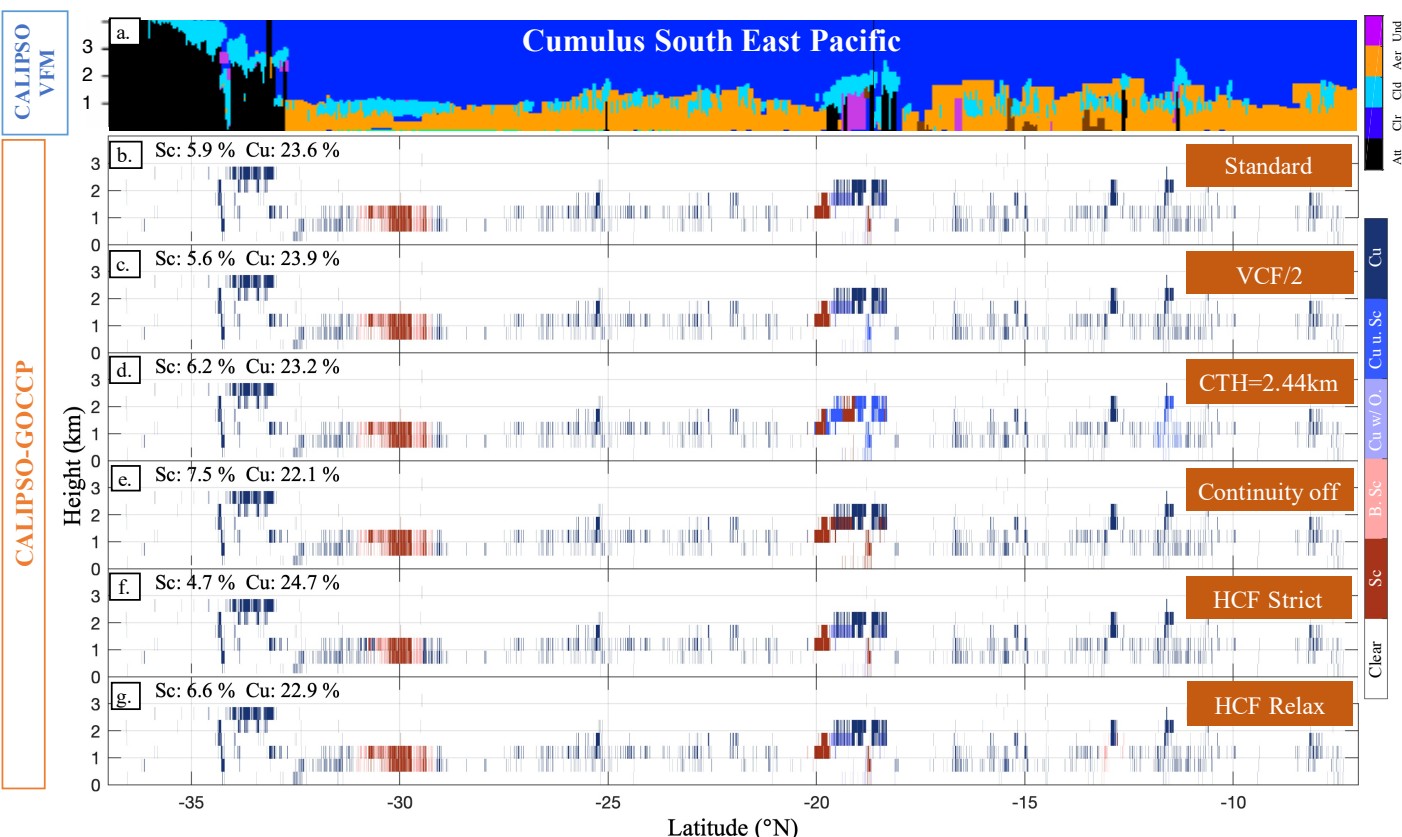

**Figure 9:** Same as Fig. 6 but for a typical cumulus case in the south-east Pacific (~ 37°S to 8°S, 2008-07-08 21:47:30, daytime).

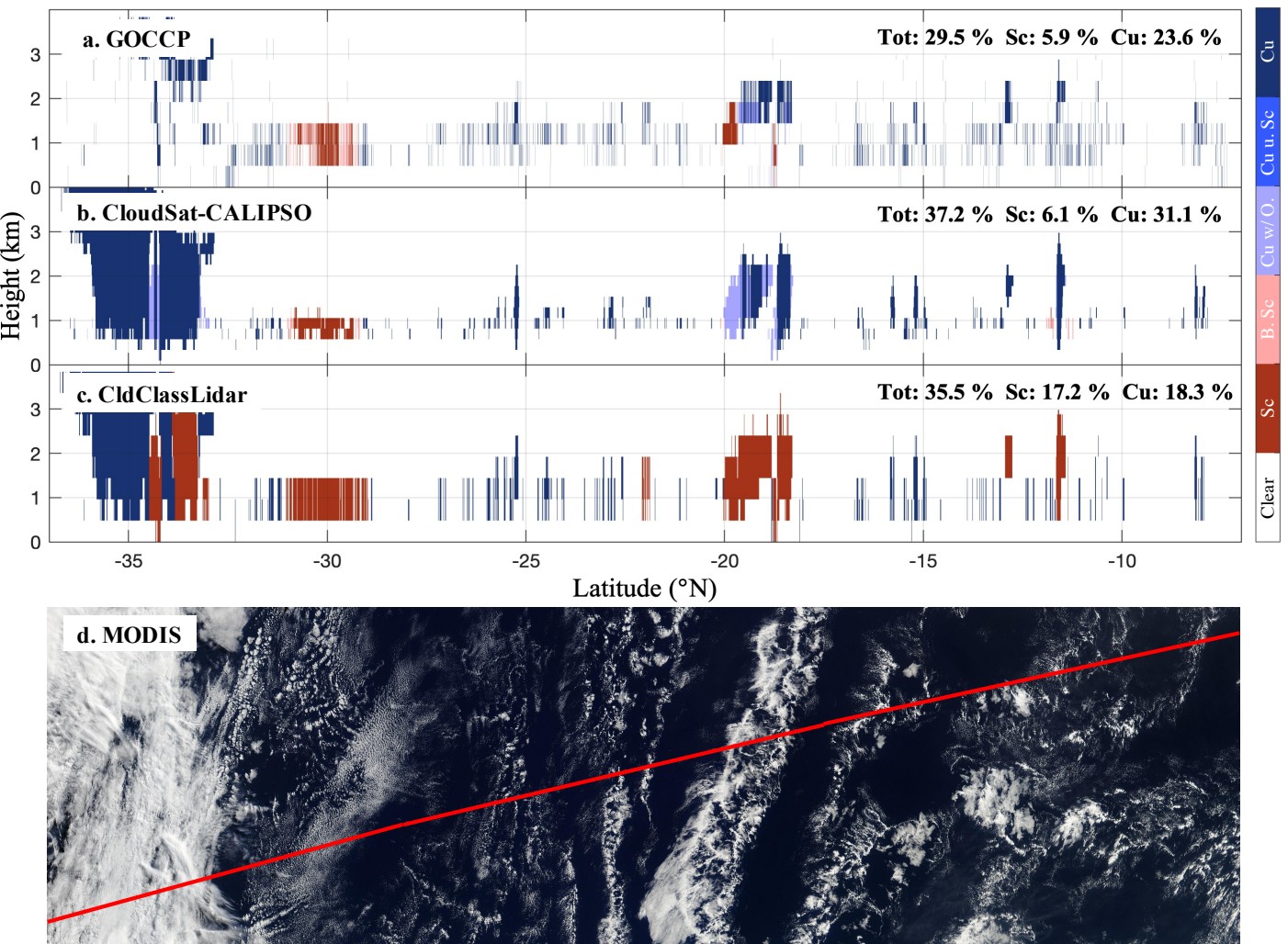

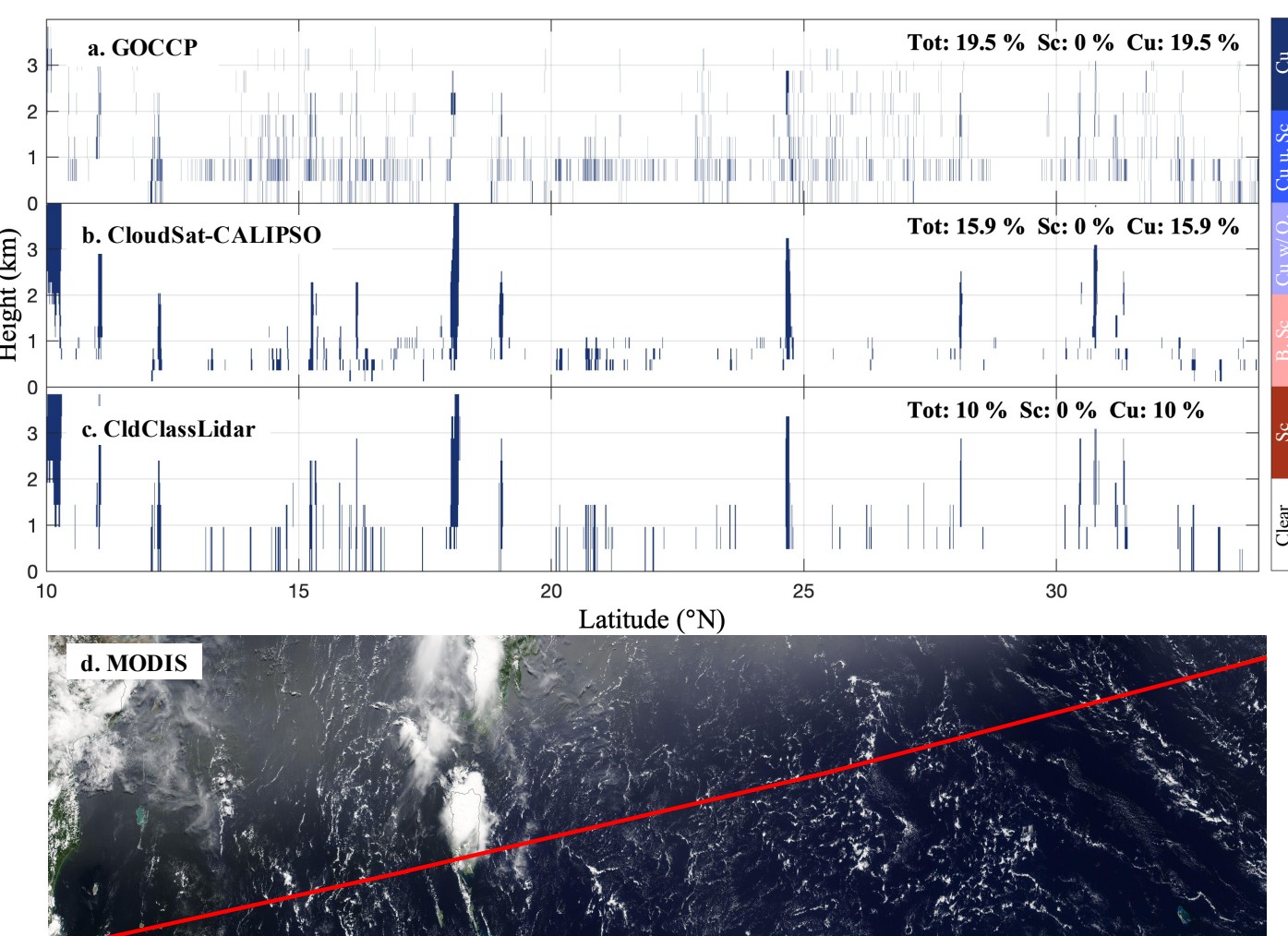

**Figure 10: Same as Fig. 8 but for another cumulus case overpassing the Barbados (~ 10°N to 35°N, 2008-07-09 17:34:08 daytime)**

**Figure 11: Same as Fig. 5 but for a typical Open-cell Sc to Cu transitioning case overlapping the south-east Pacific and the Southern Ocean (~ 55˚S to 20˚S, 2008-07-14 21:10:19, daytime).**

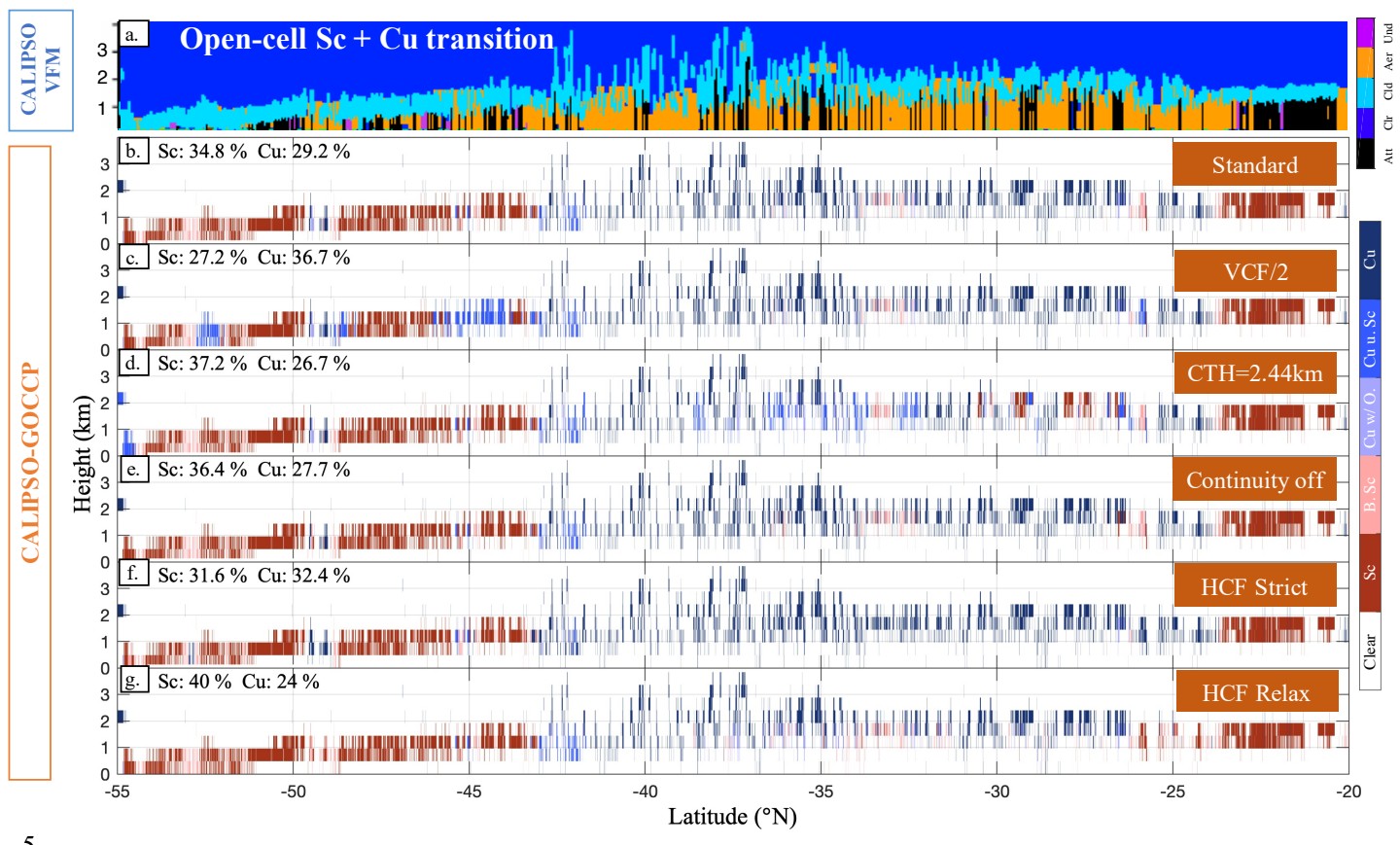

**Figure 12:** Same as Fig. 6 but for a typical Open-cell Sc to Cu transitioning case overlapping the south-east Pacific and the Southern Ocean (~ 55°S to 20°S, 2008-07-14 21:10:19, daytime).

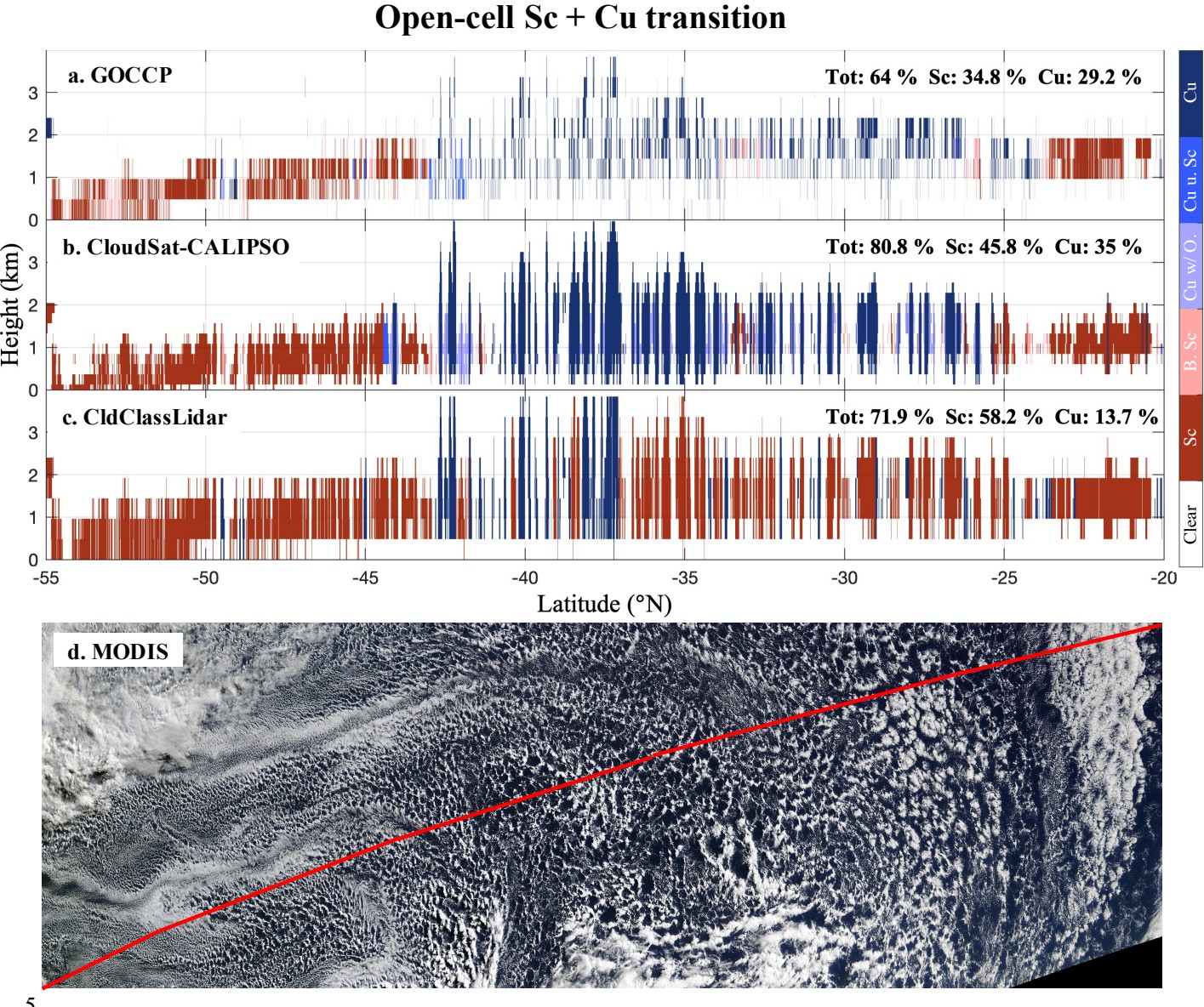

**Figure 13: Maps (x axis, longitude [˚E]; y axis, latitude [˚N]) of (top to bottom) low, Sc , Cu,  transitioning (i.e., broken Sc, Cu under Sc and Cu with stratiform outflow) and the ratio Sc to Sc-and-Cu cloud fraction (%) for (left to right) GOCCP (2007-2016), RL-GeoProf (2007-2010), 2BCCL (2007-2010), MISR (2003-2012), MODIS Terra and Aqua (2003-2015) and ISCCP (1983-2008). Different color bars are used to better separate each type of cloud. Note that for active-sensor satellites, the low category accounts for all clouds present below 3.36 km regardless of their cloud top height, hence the sum of Sc, Cu and transitioning cloud fraction can be smaller than the low cloud fraction. All datasets are averaged onto a 2.5˚x2.5˚ grid.**

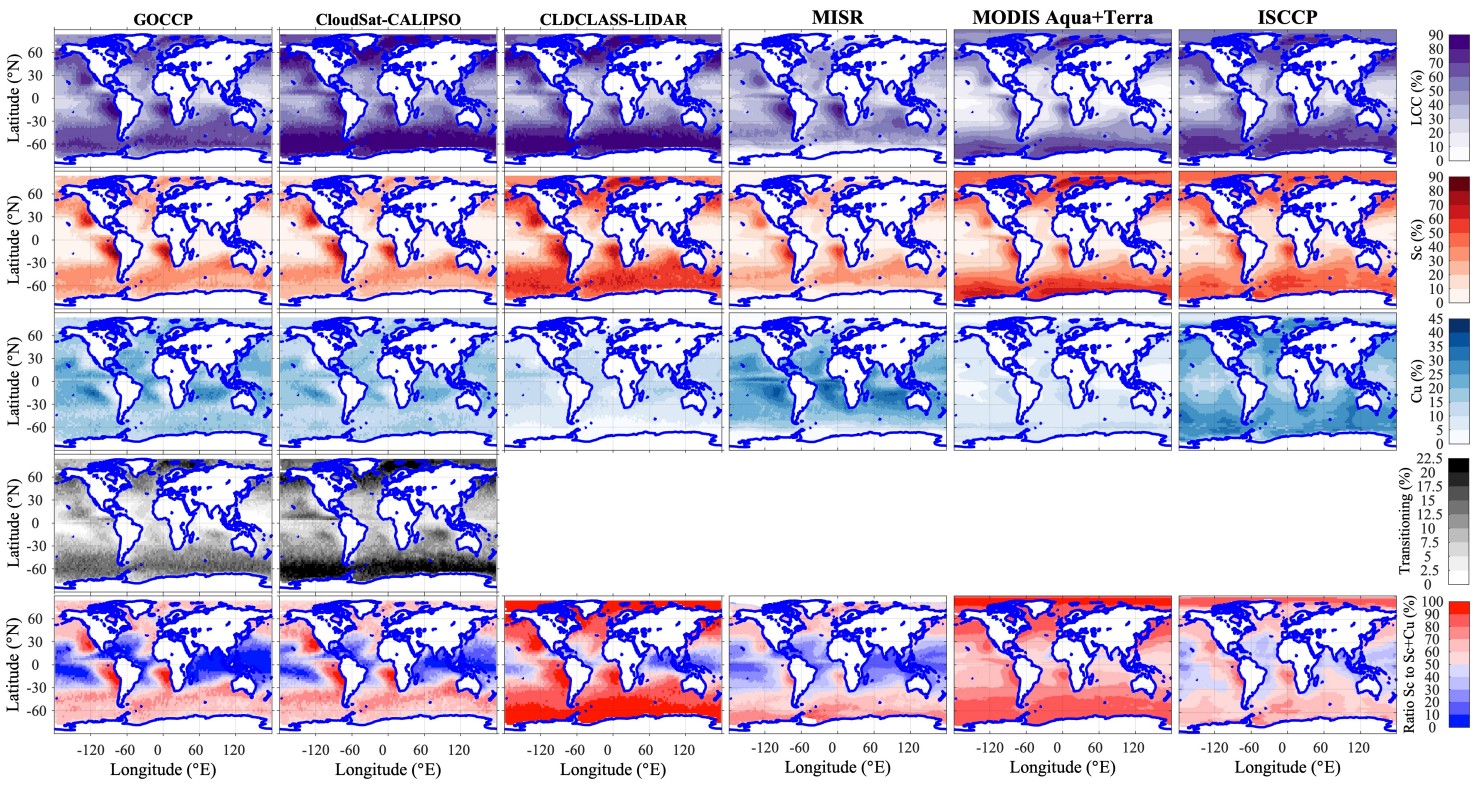

**Figure 14: Global zonal mean (%, left) and global area-weighted mean (%, right) of low, Sc, Cu, transitioning (i.e., broken Sc, Cu under Sc and Cu with stratiform outflow) and the ratio Sc to Sc-Cu clouds –from the top to the bottom– for all the satellite datasets using the same time period as in Fig. 13. Active-sensor observations are represented in solid lines as opposed to dashed lines for COT-CTP passive-sensor observations and dotted lines for clustering passive-sensor observations. The clustering area-weigthed means are represented by transparent bars with red borders. Note that the y-axis are different in every subplots. All datasets are averaged onto a 2.5˚x2.5˚ grid.**

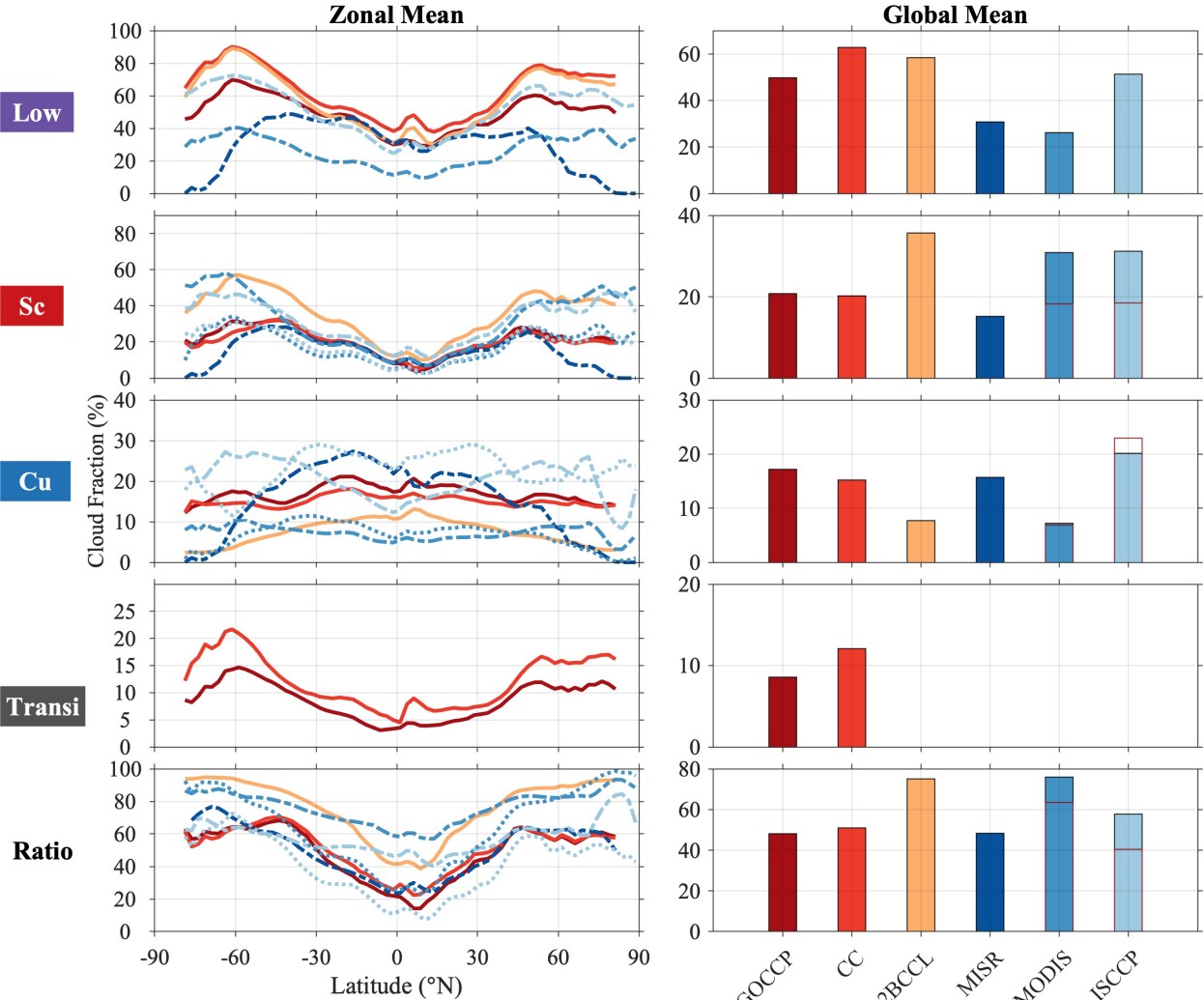

**Figure 15: Similar to Fig. 13, maps (x axis, longitude [°E]; y axis, latitude [°N]) of (top to bottom) Sc , Cu and the ratio Sc to Sc-and-Cu cloud fraction (%) for (left to right) GOCCP (2007-2016), RL-GeoProf (2007-2010), MODIS Terra and Aqua Cloud Regime (2003-2015) and ISCCP Weather State (2001-2008).**

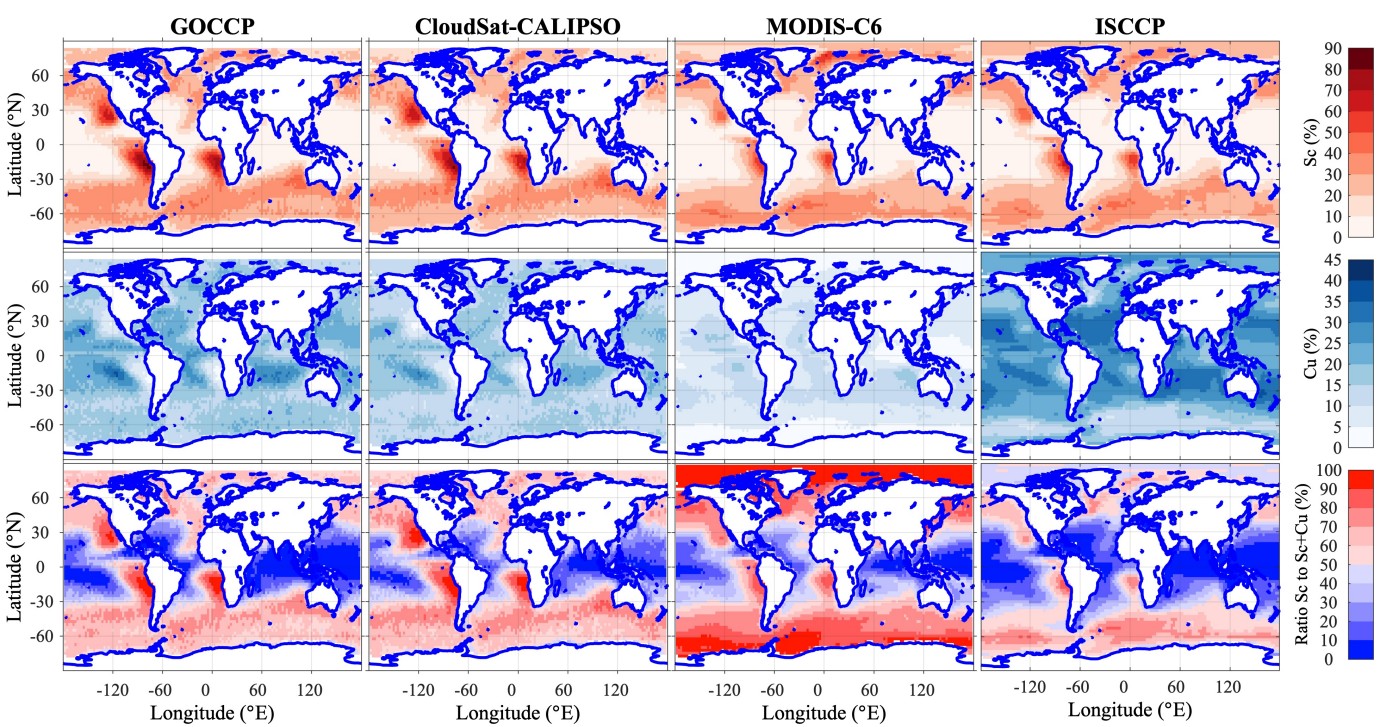

**Figure 16: Zonal profiles (x axis, latitude [°N]; y axis, height [km]) of low (first row), Sc (second row), Cu (third row) and transitioning (i.e., broken Sc, Cu under Sc and Cu with stratiform outflow; fourth row) cloud fraction (%) for GOCCP (first column), RL-GeoProf (second column) and 2BCCL (third column). As in Fig. 13, different color bars are used to better separate each type of cloud. The corresponding profiles for tropical subsidence regimes (ω500 > 10 hPa/day between 35°S/N following Cesana et al., 2019a) are represented in the fourth column for GOCCP (solid line), RL-GeoProf (dashed line) and 2BCCL (dotted line). Note that extratropical profiles are shown in Fig. S10.**

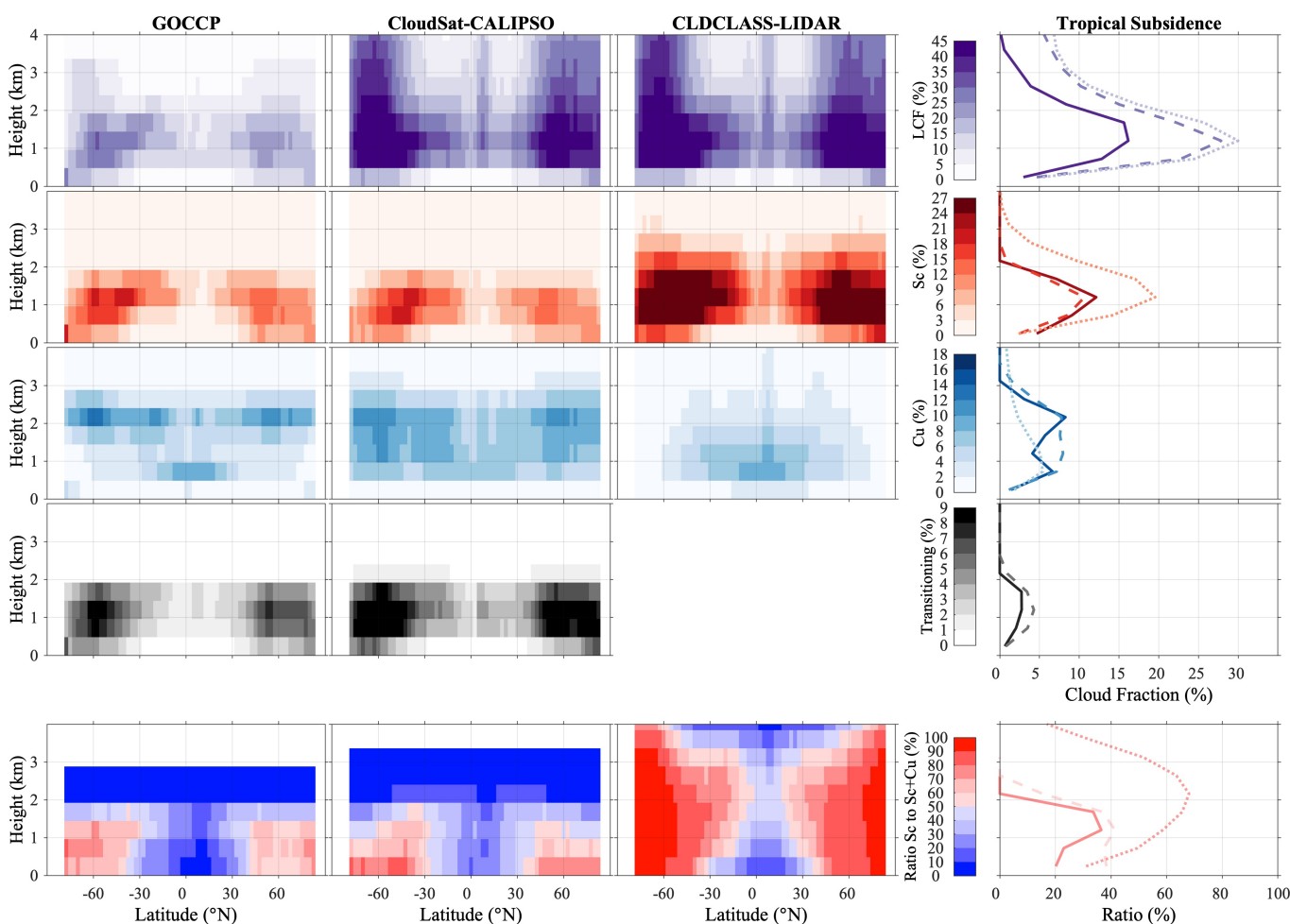