# Peer review of "The Cumulus And Stratocumulus CloudSat-CALIPSO Dataset (CASCCAD)"

_Earth System Science Data, 2019_

## Referee Comment (RC1) · Anonymous Referee #1 · 28 May 2019

Summary:

This paper describes a simple algorithm (CASCCAD) to classify low level clouds into either stratocumulus or cumulus based on cloud top height, horizontal cloud fraction, and vertical cloud fraction as measured by active space borne sensors (either lidar or lidar+radar). This algorithm is then compared against an existing radar/lidar algorithm (cldclass). The new dataset (CASCCAD) identifies far more cumulus than (cldclass), which agrees better with our a-priori assumption about the geographic distribution of these cloud types. An extremely simple classification based on the cloud optical thickness and cloud top pressure is also examined and found not to perform favorable compared to the CASCCAD dataset.

The paper is well written. It represents a novel contribution to the field. The paper

should be published following some major revisions. My main complaint with the paper is the comparisons with COT-CTP classification methodologies. For quite some time now people have been using more sophisticated k-means approaches that use a 2d joint histogram of COT-CTP to classify imager data into 'weather states'. If the authors want to compare their algorithm to the imager algorithms they should use the more recent approaches. I don't believe that the current comparison represents a fair assessment of how well imager data can be used to classify cloud regimes. Below is a limited list of papers that use MODIS based k-means regimes.

10.1002/2013JD021409, 10.1002/2016jd025193, 10.1002/2015jd024502, 10.1002/2016jd026120, 10.1007/s00382-016-3064-0, 10.1007/s00382-017-3806-7

I believe that the CASCCAD should be compared against the k-means approach ideally from MODIS but possibly also from ISCCP. The MODIS based regimes do have a clear advantage over CASCCAD; there is believable way to compare model output to those regimes. The author need to seriously elaborate on how CASCCAD can be used to evaluate models or drop the statements about model evaluation. I suggest the latter as I don't think it is possible to do a meaningful comparison. Other than these two points, I only have minor comments which are listed below.

Comments:

-Page 3, line 23: should be 1.4x1.7 km (tanelli et al., 2008). 1.1 is the distance between adjacent pixels but they overlap somewhat.

-page 4, line 31. Same as above

-page 6, line 18: The RL-GeoProf release 5 (newer version) is now available past the 2011 anomaly.

-page 7, line 18: why are attenuate profiles excluded from HCF? They are certainly cloudy right?

-page 7, section 3.4: can you explain here what influence multiple scattering has on the signal? It seems to me that when you see several bins of lidar return you are probably looking at a multiply scattered signal.

-Section 4.2.1: This COT-CTP method is quite simple. The k-means methods of Rossow et al. (2005) applied to ISCCP or more recently Oreopolous (https://doi.org/10.1002/2013JD021409) applied to MODIS probably do a much better job because they consider the joint histograms of COT-CTP over a large area. You should at least comment on this. There is a 1x1 degree daily gridded dataset that has maps of the regimes. If I were you I would get this data and plot his regimes next to yours. The aqua MODIS regimes are even coincident in time and space with yours.

-Page 10, line 29: I would remove this statement unless you want to elaborate. I can't see how you could use this data to evaluate a GCM in anything but a qualitative way. There is generally nothing in the model called Sc and Cu to compare against. Also, there is no way to apply an instrument simulator to recreate the data using the model output because the categories depend on spatial continuity.

-Figure 15: I'm concerned about the cloud fraction PDF's (left column). What is the resolution over which the cloud fraction PDF's are calculated? Is the resolution different for the different products? The PDF depends on the resolution so if the different products have different resolution the PDF's will be different just based on that fact. For this reason, the nadir-only

-sampling of the CALIPSO-CloudSat data will inherently be different than a 2D imager. I think you should remove these panels because of these concerns and I also don't think they add much to the paper.

-Page 123, line 20: Again, I don't know how you use this to evaluate a model. Just because the model has some arbitrary distinction between a boundary layer parameterization and a shallow convection parameterization doesn't mean that this is the same as stratocumulus and cumulus as you have defined them from the observations.

Furthermore, newer parameterization are beginning to 'unify' these distinct regimes (e.g. CLUBB, EDMF). I think the dataset is interesting enough in its own right without having to sell it as a model evaluation tool.

---

## Referee Comment (RC2) · Anonymous Referee #2 · 9 Jul 2019

This paper describes a new global cloud dataset built from active sensor CALIPSO and CloudSat observations. The distinguishing feature of the dataset is that cloud profile data is used to discriminate between stratocumulus and shallow cumulus, rather than relying on cloud-top altitude and optical depth derived from passive sensors. The motivation for this effort is nicely summarized in the introduction. The paper is well written and the work is a useful contribution to the field. The paper deserves publication but would benefit from some additional discussion and clarification of a few points.

1. The algorithm used to identify and discriminate cumulus and stratocumulus is described well, but the paper needs some description of the actual CASCCAD data set:

Is this a Level 2 or Level 3 dataset? What is the spatial resolution of the product? What parameters are reported – are cloud base and top altitudes included?

[Figure]

Section 5 says CASCCAD dataset is formed using both GOCCP and RL-Geoprof. How? Are these two datasets merged together somehow, or are they included as two separate elements of CASCCAD?

2. As reported here, classifications based on observed 3D structure appear to be better than classifications based on passive retrievals of cloud altitude and optical depth, which are subject to ambiguities. It would be useful to cite Mace and Wrenn (2013) who discuss a comparison in the eastern Pacific

Mace and Wrenn, 2013: Evaluation of the Hydrometeor Layers in the East and West Pacific within ISCCP Cloud Top Pressure-Optical Depth Bins Using Merged CloudSat and CALIPSO Data, J. Climate, doi:10.1175/JCLI-D-12-00207.1

3. Referee #1 dismisses the utility of CASCCAD for model evaluation for reasons which are not clear. CASCCAD appears quite useful for model evaluation to me. Some additional discussion of this would be useful, especially to clarify differences in using GOCCP (which is based on a simulator approach) versus RL-Geoprof (which is not).

4. Finally, I agree with Referee #1 that it would be interesting and useful for the authors to say something about the weaknesses and strengths of CASCCAD vs. weather state approaches to analyzing passive observations.

Minor points:

The last sentence of the first paragraph of the Introduction repeats an earlier sentence.

To document limitations of CloudSat in sensing shallow clouds it would be useful to cite: Liu, D., Q. Liu, L. Qi and Y. Fu, 2016: "Oceanic single-layer warm clouds missed by the Cloud Profiling Radar as inferred from MODIS and CALIOP measurements", JGR, doi:10.1002/2016JD025485.

"CALIPSO" is mis-spelled in the caption of Figure 6

---

## Author Comment (AC1) · 6 Sep 2019

The authors are very grateful for the helpful comments provided by both reviewers. Our point-by-point responses are provided below in blue directly after the reviewers' comments.

To summarize, we made 5 main changes in the manuscript:
1) we now use the latest 2b-cldclass-lidar release (R05) in the case studies and statistics since the changes compared to R04 are substantial –although the conclusions remain unchanged–, which is not the case for the RL-GeoProf (reviewer 1).
2) we further justify future use of CASCCAD dataset for model evaluation in the conclusions (reviewer 1).
3) we now compare the CASCCAD datasets with the passive-sensor clustering datasets in addition to the COT-CTP technique (reviewer 1 and 2).
4) we further define the content of the CASCCAD dataset at the beginning of the dataset section (reviewer 2).
5) finally, we slightly modified the CASCCAD algorithm to keep track of the cloud type called Cu with stratiform outflow, which was accounted for as pure Cu before. As a result:
   a. the algorithm chart flow (fig. 2) has been updated as well as the cartoon explaining the VCF and horizontal continuity test logic (Fig. 5).
   b. the case studies have been updated although the changes are minor
   c. the global statistics are slightly changed in RL-GeoProf CASCCAD (a little less/more Cu/transitioning cloud) while it is negligible in GOCCP-CASCCAD.
   d. these changes do not impact the conclusions of the study

Reviewer 1
Summary
This paper describes a simple algorithm (CASCCAD) to classify low level clouds into either stratocumulus or cumulus based on cloud top height, horizontal cloud fraction, and vertical cloud fraction as measured by active space borne sensors (either lidar or lidar+radar). This algorithm is then compared against an existing radar/lidar algorithm (cldclass). The new dataset (CASCCAD) identifies far more cumulus than (cldclass), which agrees better with our a-priori assumption about the geographic distribution of these cloud types. An extremely simple classification based on the cloud optical thickness and cloud top pressure is also examined and found not to perform favorable compared to the CASCCAD dataset.
The paper is well written. It represents a novel contribution to the field. The paper should be published following some major revisions.

My main complaint with the paper is the comparisons with COT-CTP classification methodologies. For quite some time now people have been using more sophisticated k-means approaches that use a 2d joint histogram of COT-CTP to classify imager data into 'weather states'. If the authors want to compare their algorithm to the imager algorithms they should use the more recent approaches. I don't believe that the current comparison represents a fair assessment of how well imager data can be used to classify cloud regimes. Below is a limited list of papers that use MODIS based k-means regimes. 10.1002/2013JD021409, 10.1002/2016jd025193, 10.1002/2015jd024502, 10.1002/2016jd026120, 10.1007/s00382-016-3064-0, 10.1007/s00382-017-3806-7. I believe that the CASCCAD should be compared against the k-means approach ideally from MODIS but possibly also from ISCCP. The MODIS based regimes do have a clear advantage over CASCCAD; there is believable way to compare model output to those regimes.

The author need to seriously elaborate on how CASCCAD can be used to evaluate models or drop the statements about model evaluation. I suggest the latter as I don't think it is possible to do a meaningful comparison.

Regarding the reviewer's two main comments: (1) k-means approaches can only indirectly be compared to CASSCAD because they do not identify individual clouds, but we have now done what we feel we can and included it in the revised manuscript. (2) We feel that CASSCAD is actually a better dataset for evaluating models in a way that is meaningful for tracing errors to their source rather than just concluding that the model is wrong, which we now discuss in the revised paper. Since the reviewer also mentions his two main concerns in the minor concerns below, we address both of these points in more detail directly below.

Other than these two points, I only have minor comments which are listed below.

Comments:
-Page 3, line 23: should be 1.4x1.7 km (tanelli et al., 2008). 1.1 is the distance between adjacent pixels but they overlap somewhat.

-page 4, line 31. Same as above
Thank you for pointing this out. We corrected this in both places in the new manuscript.

-page 6, line 18: The RL-GeoProf release 5 (newer version) is now available past the 2011 anomaly.
The new version came out after we submitted the study. However, even with the additional R05 time period, a substantial amount of data are still missing, which would still limit the possibility of computing statistical relationships between cloud amount and environmental variables. We reproduced the case studies with the R05 version of RL-GeoProf and we found that the change is negligible (see new Fig. S2-3-4-5). For these reasons, we decided not to update the results with the R05 for the sake of computational, space and time resources. We now explain this at the end of section 3.1: *"Note that the release R05 of RL-GeoProf came out after the submission of the original manuscript (late May 2019), which includes data after April 2011. However, these are for daytime only and substantial periods of time are still missing (e.g., May 2011 through May 2012 and the whole year 2014), which makes it difficult to compute a consistent climatology and derive statistical relationships between clouds and environmental variables. Additionally, the differences between CASCCAD using RL-GeoProf R04 and R05 are very small in the case studies of section 4.1 (see Fig. S2-S3-S4-S5). There is a small decrease of the overall cloud fraction (76.3 vs. 75.8 %, 37.2 vs. 35.2 %, 15.9 vs. 16 % and 80.8 vs. 77.2 % for R04 and R05 respectively), which affects mostly the Cu cloud fraction. Since the change is almost negligible, we decided not to update the global statistics with the R05 version for the sake of computational, space and time resources."*

Concerning the 2BCCL product, the change between R04 and R05 is substantial. The overall excess of low-level cloud fraction over the tropical ocean is somewhat fixed in R05, likely due to the use of a different lidar product (LIDAR-AUX, Wang et al., 2013). However, the excess and lack of Sc and Cu, respectively, are still present, most likely because there is no or little change in the 2BCCL algorithm for Sc, St and Cu clouds in the low levels (http://www.cloudsat.cira.colostate.edu/sites/default/files/products/files/2B-CLDCLASS-LIDAR_PDICD.P1_R05.rev0_.pdf).
It is mentioned in section 2.3 that we use R05 version and that the reader should refer to the original version of the manuscript for results using the older R04: *"Note that there are substantial differences between results using the R04 and R05 versions, which is why the reader should refer to the original manuscript published in May, 22nd 2019 on the ESSD discussion website for results using the older R04 version."*

-page 7, line 18: why are attenuate profiles excluded from HCF? They are certainly cloudy right?

The way attenuated pixels are treated is consistent with what is done in the GOCCP product for cloud fraction computations. Here, we exclude the profile when all pixels (= 4 pixels) below 1.92 km (which is where the HCF is computed) are fully attenuated pixels because it is not possible to know with certainty whether these pixels are clear or cloudy.

We modified this sentence to better explain why this is done: *"Finally, note that when all 480-m-pixels below 1.92 km are fully attenuated (4 pixels), the profile is excluded from the HCF computation – consistent with what is done in the GOCCP product for cloud fraction computations– because we do not know whether these pixels are clear or cloudy."*

-page 7, section 3.4: can you explain here what influence multiple scattering has on the signal? It seems to me that when you see several bins of lidar return you are probably looking at a multiply scattered signal.

The CASCCAD algorithm is applied to level 2 GOCCP and CloudSat-CALIPSO files, which already include orbital cloud mask profiles. Therefore, we do not use directly the attenuated backscattered signal from the lidar, we simply compute an along-track cloud fraction using the GOCCP or CloudSat-CALIPSO cloud mask, which both take into account the multiple scattering issues in their algorithms.

-Section 4.2.1: This COT-CTP method is quite simple. The k-means methods of Rossow et al. (2005) applied to ISCCP or more recently Oreopolous (https://doi.org/10.1002/2013JD021409) applied to MODIS probably do a much better job because they consider the joint histograms of COT-CTP over a large area. You should at least comment on this. There is a 1x1 degree daily gridded dataset that has maps of the regimes. If I were you I would get this data and plot his regimes next to yours. The aqua MODIS regimes are even coincident in time and space with yours.

The WS and CR approaches do not discriminate cloud type or even low cloud from middle and high clouds. They represent mixtures of cloud types although one cloud type is often prevalent. As such, it is not exactly consistent to directly compare them with the CASCCAD datasets, however, we agree with the reviewers that showing the CR and WS more recent approaches would provide a broader context and additional information. For this reason, we added the ISCCP WSs and MODIS CRs observations in section 4 of the analysis although we already briefly mentioned these techniques in the first version of the manuscript (p11 line 9).

There is an additional figure (new Figure 15) with the ISCCP WSs and MODIS CRs geographical distributions and we added the zonal and global means to the existing Figure 14. In addition, we discuss the new results in section 4.2.1. In essence, the clustering-derived datasets show a better agreement with CALIPSO and CloudSat-CALIPSO CASCCAD particularly in the tropics and for the Sc clouds. The Cu-regimes are better correlated with CALIPSO CASCCAD in terms of geographical distribution although MODIS underestimates their fraction while ISCCP overestimates it (likely due to mid and high-level clouds contained in WS7 and WS8).

-Page 10, line 29: I would remove this statement unless you want to elaborate. I can't see how you could use this data to evaluate a GCM in anything but a qualitative way. There is generally nothing in the model called Sc and Cu to compare against. Also, there is no way to apply an instrument simulator to recreate the data using the model output because the categories depend on spatial continuity.

We disagree with the reviewer's comments. Models make Sc and Cu explicitly, based on specific but different physical mechanisms. We understand that some models are now using unified turbulence schemes of one type or another, but these schemes still have to produce Cu and Sc and the transition between them under appropriate conditions. We discuss in response to a later comment why CASSCAD is valuable for the evaluation of such parameterizations. We also note that Reviewer 2 feels that CASSCAD is useful for model evaluation.

It is true that it would be difficult to recreate the same Cu Sc diagnostic in the simulator as in the CASCCAD algorithm, however:

1) The simulator sums up the convective and stratiform cloud fraction before computing the diagnostics and therefore one could just separate their contribution rather than summing them up.

2) A simulator is not necessarily needed for this particular type of comparison, because we identify the different cloud modes explicitly and we can select regimes in which lidar attenuation is negligible (e.g., ω500 > 0 hPa/day, Cesana et al., 2019a)

We expanded the last paragraph of the conclusion to discuss this: "*Finally, one of the reasons we developed CASCCAD is to provide an improved observational constraint for low-level cloud feedbacks in GCMs. Although the CASCCAD DA cannot be implemented in a lidar simulator (Chepfer et al., 2008), it is still possible to use CASCCAD datasets for model evaluation because i) both the convective and stratiform cloud fraction are provided as inputs to the lidar simulator and could be easily saved separately rather than summed up; and ii) a simulator is not necessarily needed for model-to-obs comparison of Cu and Sc clouds over the tropical oceans, because we identify the different cloud modes explicitly and we can select regimes in which lidar attenuation is negligible (e.g., $\omega_{500} > 0$ hPa/day, Cesana et al., 2019a).*"*

-Figure 15: I'm concerned about the cloud fraction PDF's (left column). What is the resolution over which the cloud fraction PDF's are calculated? Is the resolution different for the different products? The PDF depends on the resolution so if the different products have different resolution the PDF's will be different just based on that fact. For this reason, the nadir-only-sampling of the CALIPSO-CloudSat data will inherently be different than a 2D imager.I think you should remove these panels because of these concerns and I also don't think they add much to the paper.

To answer the reviewer's concerns, we removed these panels. We also averaged MISR and MODIS onto a 2.5x2.5 grid, similar to the other products, in the new version of this figure. We now mention this information in the figure's caption.

-Page 123, line 20: Again, I don't know how you use this to evaluate a model. Just because the model has some arbitrary distinction between a boundary layer parameterization and a shallow convection parameterization doesn't mean that this is the same as stratocumulus and cumulus as you have defined them from the observations. Furthermore, newer parameterization are beginning to 'unify' these distinct regimes (e.g. CLUBB, EDMF). I think the dataset is interesting enough in its own right without having to sell it as a model evaluation tool.

Here we also disagree with the reviewer's comment. As stated above, the models do actually simulate Sc and Cu based on physical mechanisms. Existing GCMs fall into two classes: (1) Those that use separate parameterizations for cumulus and stratocumulus clouds, for which CASSCAD is directly applicable as an evaluation tool to determine whether biases are attributable to process assumptions made about one cloud type vs. the other. (2) Those that use unified turbulence parameterizations that are intended to represent the full spectrum of boundary layer clouds. Even for the latter class of parameterizations, though, the models are in effect capturing the two cloud types and the transition between them via different segments of the pdf and making assumptions about when each type of cloud is active, either in the physics itself or in the diagnosis of the model behavior. In both cases, the distinction between Cu and Sc is made crudely using criteria such as inversion strength or 500 mb vertical velocity as the "definition" of one vs. the other cloud type. For example, Koehler et al. (2011, QJRMS), in an EDMF scheme, use inversion strength as a "decoupling criterion" to determine when they should turn off the shallow convection (MF) component of EDMF. Bogenschutz et al. (2013, J. Climate) use inversion strength and vertical velocity to define "stratocumulus," "transition," and "cumulus" regimes for the purpose of understanding where CLUBB is performing well vs. poorly. We feel that a dataset that directly diagnoses the clouds themselves is ultimately a more reliable indicator than assumptions about cloud types based on

large scale environmental properties.  CASSCAD in fact allows the relationship between Sc, Cu, and environmental state to be diagnosed directly rather than being assumed. Given the large impact that schemes like CLUBB have on climate sensitivity (Gettelman et al. 2019, GRL), there is an increasing need to use metrics beyond mean state biases to decide what is and is not realistic.   Note that we did not explain this in the previous manuscript. We now specify this in the description of the algorithm, section 3.1: *"However, one cannot separate clouds according to the mechanisms that form them as GCMs do using different PBL and convective parameterizations, which is why we choose to use the morphology to discriminate cloud types in this study."*

Based on our study, it seems that CASCCAD is doing better at identifying Cu-Sc clouds than the other methods –while it is not perfect– and therefore CASCCAD looks like a good candidate for model evaluation.

We now specify this in the last paragraph of the conclusion: *"Finally, ... to judge model realism and fidelity"*.

---

## Author Comment (AC2) · 6 Sep 2019

The authors are very grateful for the helpful comments provided by both reviewers. Our point-by-point responses are provided below in blue directly after the reviewers' comments.

To summarize, we made 5 main changes in the manuscript:
1) we now use the latest 2b-cldclass-lidar release (R05) in the case studies and statistics since the changes compared to R04 are substantial –although the conclusions remain unchanged–, which is not the case for the RL-GeoProf (reviewer 1).
2) we further justify future use of CASCCAD dataset for model evaluation in the conclusions (reviewer 1).
3) we now compare the CASCCAD datasets with the passive-sensor clustering datasets in addition to the COT-CTP technique (reviewer 1 and 2).
4) we further define the content of the CASCCAD dataset at the beginning of the dataset section (reviewer 2).
5) finally, we slightly modified the CASCCAD algorithm to keep track of the cloud type called Cu with stratiform outflow, which was accounted for as pure Cu before. As a result:
   a. the algorithm chart flow (fig. 2) has been updated as well as the cartoon explaining the VCF and horizontal continuity test logic (Fig. 5).
   b. the case studies have been updated although the changes are minor
   c. the global statistics are slightly changed in RL-GeoProf CASCCAD (a little less/more Cu/transitioning cloud) while it is negligible in GOCCP-CASCCAD.
   d. these changes do not impact the conclusions of the study

Reviewer 2
This paper describes a new global cloud dataset built from active sensor CALIPSO and CloudSat observations. The distinguishing feature of the dataset is that cloud profile data is used to discriminate between stratocumulus and shallow cumulus, rather than relying on cloud-top altitude and optical depth derived from passive sensors. The motivation for this effort is nicely summarized in the introduction. The paper is well written and the work is a useful contribution to the field. The paper deserves publication but would benefit from some additional discussion and clarification of a few points.

1. The algorithm used to identify and discriminate cumulus and stratocumulus is described well, but the paper needs some description of the actual CASCCAD data set:
Is this a Level 2 or Level 3 dataset? What is the spatial resolution of the product? What parameters are reported – are cloud base and top altitudes included?
Section 5 says CASCCAD dataset is formed using both GOCCP and RL-Geoprof. How? Are these two datasets merged together somehow, or are they included as two separate elements of CASCCAD?
We acknowledge that the description of the dataset was somewhat ambiguous in the previous manuscript. The CASCCAD datasets are actually two distinct datasets using the same algorithm but applied to two different level 2 observational datasets, which are CALIPSO-GOCCP and RL-GeoProf. There are level 2 and level 3 CASCCAD files for both datasets. The level 2 files use the native resolution of the GOCCP and RL-GeoProf datasets (i.e., 333 m x 480 m for GOCCP and ~1.1 km x 240 m for RL-GeoProf). The level 3 files are global statistics over a 2.5°x2.5° grid over 40 480 m levels for each month between 2007 and 2016, which are then averaged over the time dimension. The monthly means will be made available once the manuscript is accepted. The parameters reported are 2D and 3D cloud fractions of low, Cu, Sc and transitioning clouds for level 3 files and cloud mask profiles of Cu, Sc and transitioning cloud types for the level 2 files. While the cloud top/base is not reported explicitly as a variable for each profile, it can be derived from the cloud mask contained in level 2 files.

We added a full paragraph at the beginning of section 2 –before section 2.1– to address the reviewer's concerns and describe what we report above. Additionally, we make it clear throughout the manuscript that the GOCCP and RL-GeoProf CASCCAD datasets are two distinct datasets based on the same algorithm but using GOCCP native level 2 observations and the combined CloudSat-CALIPSO RL-GeoProf level 2 observations, respectively. For example, in the last paragraph of the introduction, the first of section 4.1 and the first paragraph of section 5, we now specify that the algorithm is apply "separately" to both GOCCP and RL-GeoProf observations. Finally, in the conclusion, when the CASCCAD datasets are mentioned we now refer to "distinct GOCCP and RL-GeoProf CASCCAD datasets" rather than only "CASCCAD datasets".

2. As reported here, classifications based on observed 3D structure appear to be better than classifications based on passive retrievals of cloud altitude and optical depth, which are subject to ambiguities. It would be useful to cite Mace and Wrenn (2013) who discuss a comparison in the eastern Pacific
Mace and Wrenn, 2013: Evaluation of the Hydrometeor Layers in the East and West Pacific within ISCCP Cloud Top Pressure-Optical Depth Bins Using Merged CloudSat and CALIPSO Data, J. Climate, doi:10.1175/JCLI-D-12-00207.1
Thank you for pointing out this relevant reference. We added it to the manuscript at the beginning of section 4.2.1 where the comparison between passive and active is performed:
*In addition, (Mace and Wrenn, 2013) showed that except for thin cirrus and Sc cloud types, the COT-derived cloud types are mostly mixtures of different cloud types in two regions of the Eastern Pacific.*

3. Referee #1 dismisses the utility of CASCCAD for model evaluation for reasons which are not clear. CASCCAD appears quite useful for model evaluation to me. Some additional discussion of this would be useful, especially to clarify differences in using GOCCP (which is based on a simulator approach) versus RL-Geoprof (which is not).
We agree with reviewer 2 and we added some additional discussion –as requested by reviewer 2– to better explain how we believe CASCCAD can be used for model evaluation in the conclusion. A point by point answer is given to reviewer 1 above (however, we added our answers to reviewer 1 with respect to the use of CASCCAD for model evaluation at the end of this response for reviewer's 2 convenience). Based on the case studies, profile and zonal means presented in the paper, the biggest differences between CASSCAD versions for the two datasets mentioned above are the tendency for the RL-GeoProf version to diagnose more transitioning clouds than the GOCCP version poleward of 40°, because i) RL-GeoProf is less affected by overlapping mid- and high-cloud attenuation and better captures the full vertical extent of low clouds and ii) RL-GeoProf contains larger cloud clusters than GOCCP, making the continuity test more efficient . We now mention this in the second paragraph of section 4.2.1.

4. Finally, I agree with Referee #1 that it would be interesting and useful for the authors to say something about the weaknesses and strengths of CASCCAD vs. weather state approaches to analyzing passive observations.
The WS and CR approaches do not exactly classify clouds by cloud type or by low, middle and high layers although they provide CTP-tau histograms for each WS/CR. Therefore they represent mixtures of cloud types among which one cloud type is often prevalent. As such it is not exactly consistent to directly compare with the CASCCAD datasets, however, we agree with the reviewers that showing the CR and WS more recent approaches would provide a broader context and additional information as their horizontal and time sampling and their time record are better than CloudSat-CALIPSO.
For this reason, we added the ISCCP WSs and MODIS CRs observations in section 4 of the analysis.

Minor points:

The last sentence of the first paragraph of the Introduction repeats an earlier sentence.
The first occurrence of this sentence has been removed.

To document limitations of CloudSat in sensing shallow clouds it would be useful to
cite: Liu, D., Q. Liu, L. Qi and Y. Fu, 2016: "Oceanic single-layer warm clouds missed
by the Cloud Profiling Radar as inferred from MODIS and CALIOP measurements",
JGR, doi:10.1002/2016JD025485.
Thank you for bringing this reference to our attention. We modified the sentence at L26 of P3 to
include that reference: "*Although the radar-only product extends over a longer time-period (for
daytime only, see section 2.2), the CPR is less sensitive to fractionated and thin shallow cumulus
clouds than the CALIPSO lidar and its ground clutter prevents cloud detection below 1 km,
which preclude the detection of a large amount of marine low-level clouds (mostly Sc and Cu
clouds, Liu et al., 2016).*".

"CALIPSO" is mis-spelled in the caption of Figure 6:
This was corrected

**Responses to reviewer 1 with respect to model evaluation using CASCCAD:**

-Page 10, line 29: I would remove this statement unless you want to elaborate. I can't
see how you could use this data to evaluate a GCM in anything but a qualitative way.
There is generally nothing in the model called Sc and Cu to compare against. Also,
there is no way to apply an instrument simulator to recreate the data using the model
output because the categories depend on spatial continuity.
We disagree with the reviewer's comments. Models make Sc and Cu explicitly, based on specific but
different physical mechanisms. We understand that some models are now using unified turbulence
schemes of one type or another, but these schemes still have to produce Cu and Sc and the transition
between them under appropriate conditions. We discuss in response to a later comment why CASSCAD
is valuable for the evaluation of such parameterizations. We also note that Reviewer 2 feels that
CASSCAD is useful for model evaluation.
It is true that it would be difficult to recreate the same Cu Sc diagnostic in the simulator as in the
CASCCAD algorithm, however:
1) The simulator sums up the convective and stratiform cloud fraction before computing the
   diagnostics and therefore one could just separate their contribution rather than summing them up.
2) A simulator is not necessarily needed for this particular type of comparison, because we identify
   the different cloud modes explicitly and we can select regimes in which lidar attenuation is
   negligible (e.g., $\omega 500 > 0$ hPa/day, Cesana et al., 2019a)
We expanded the last paragraph of the conclusion to discuss this: "*Finally, one of the reasons we
developed CASCCAD is to provide an improved observational constraint for low-level cloud feedbacks in
GCMs. Although the CASCCAD DA cannot be implemented in a lidar simulator (Chepfer et al., 2008), it
is still possible to use CASCCAD datasets for model evaluation because i) both the convective and
stratiform cloud fraction are provided as inputs to the lidar simulator and could be easily saved
separately rather than summed up; and ii) a simulator is not necessarily needed for model-to-obs
comparison of Cu and Sc clouds over the tropical oceans, because we identify the different cloud modes
explicitly and we can select regimes in which lidar attenuation is negligible (e.g., $\omega_{500} > 0$ hPa/day,
Cesana et al., 2019a).*"

-Page 123, line 20: Again, I don't know how you use this to evaluate a model. Just because the model has some arbitrary distinction between a boundary layer parameterization and a shallow convection parameterization doesn't mean that this is the same as stratocumulus and cumulus as you have defined them from the observations. Furthermore, newer parameterization are beginning to 'unify' these distinct regimes (e.g. CLUBB, EDMF). I think the dataset is interesting enough in its own right without having to sell it as a model evaluation tool.

Here we also disagree with the reviewer's comment. As stated above, the models do actually simulate Sc and Cu based on physical mechanisms. Existing GCMs fall into two classes: (1) Those that use separate parameterizations for cumulus and stratocumulus clouds, for which CASSCAD is directly applicable as an evaluation tool to determine whether biases are attributable to process assumptions made about one cloud type vs. the other. (2) Those that use unified turbulence parameterizations that are intended to represent the full spectrum of boundary layer clouds. Even for the latter class of parameterizations, though, the models are in effect capturing the two cloud types and the transition between them via different segments of the pdf and making assumptions about when each type of cloud is active, either in the physics itself or in the diagnosis of the model behavior. In both cases, the distinction between Cu and Sc is made crudely using criteria such as inversion strength or 500 mb vertical velocity as the "definition" of one vs. the other cloud type. For example, Koehler et al. (2011, QJRMS), in an EDMF scheme, use inversion strength as a "decoupling criterion" to determine when they should turn off the shallow convection (MF) component of EDMF. Bogenschutz et al. (2013, J. Climate) use inversion strength and vertical velocity to define "stratocumulus," "transition," and "cumulus" regimes for the purpose of understanding where CLUBB is performing well vs. poorly. We feel that a dataset that directly diagnoses the clouds themselves is ultimately a more reliable indicator than assumptions about cloud types based on large scale environmental properties. CASSCAD in fact allows the relationship between Sc, Cu, and environmental state to be diagnosed directly rather than being assumed. Given the large impact that schemes like CLUBB have on climate sensitivity (Gettelman et al. 2019, GRL), there is an increasing need to use metrics beyond mean state biases to decide what is and is not realistic. Note that we did not explain this in the previous manuscript. We now specify this in the description of the algorithm, section 3.1: *"However, one cannot separate clouds according to the mechanisms that form them as GCMs do using different PBL and convective parameterizations, which is why we choose to use the morphology to discriminate cloud types in this study."*

Based on our study, it seems that CASCCAD is doing better at identifying Cu-Sc clouds than the other methods –while it is not perfect– and therefore CASCCAD looks like a good candidate for model evaluation.

We now specify this in the last paragraph of the conclusion: *"Finally, ... to judge model realism and fidelity".*